# The non-redundant functions of PIWI family proteins in gametogenesis in golden hamsters

Xiaolong Lv [1,3], Wen Xiao[1,3], Yana Lai[2,3], Zhaozhen Zhang[2,3], Hongdao Zhang [1,3], Chen Qiu[2], Li Hou[1], Qin Chen[2], Duanduan Wang[1], Yun Gao[2], Yuanyuan Song[1], Xinjia Shui[1], Qinghua Chen[2], Ruixin Qin[2], Shuang Liang[2], Wentao Zeng [2], Aimin Shi[2] ✉, Jianmin Li [2] ✉ & Ligang Wu [1] ✉

The piRNA pathway is essential for female fertility in golden hamsters and likely humans, but not in mice. However, the role of individual PIWIs in mammalian reproduction remains poorly understood outside of mice. Here, we describe the expression profiles, subcellular localization, and knockout-associated reproductive defects for all four PIWIs in golden hamsters. In female golden hamsters, PIWIL1 and PIWIL3 are highly expressed throughout oogenesis and early embryogenesis, while knockout of PIWIL1 leads to sterility, and PIWIL3 deficiency results in subfertility with lagging zygotic development. PIWIL1 can partially compensate for TE silencing in PIWIL3 knockout females, but not vice versa. PIWIL1 and PIWIL4 are the predominant PIWIs expressed in adult and postnatal testes, respectively, while PIWIL2 is present at both stages. Loss of any PIWI expressed in testes leads to sterility and severe but distinct spermatogenesis disorders. These findings illustrate the non-redundant regulatory functions of PIWI-piRNAs in gametogenesis and early embryogenesis in golden hamsters, facilitating study of their role in human fertility.

PIWI-interacting RNAs (piRNAs) are a class of 18-35 nucleotide small noncoding RNAs that associate with PIWI proteins to form a germline-specific effector complex known as the piRNA-induced silencing complex (piRISC)[1–10]. The piRISC plays multiple roles in silencing transposable elements (TEs), regulating protein-coding genes, defending against viral infection, and is essential for fertility[10–13].

The biological functions of mammalian *Piwi* genes are mainly investigated in mouse models[14–17]. Previous studies in mice led to the assumption that *Piwi* genes are essential for male fertility but dispensable for female fertility. However, only *Piwil1*, *Piwil2*, and *Piwil4* are encoded in the mouse genome, although *Piwil3* is present in most other mammalian genomes[8,9,18,19]. Moreover, mouse oocytes produce an abundance of endogenous small interfering RNAs (endo-siRNAs)[20–22], while oocytes of other mammals do not[7–9]. These observations suggest that mice may be an exception to the general principles of small RNA composition in the germ cells of mammals. Recently, three independent studies in golden hamsters (*Mesocricetus auratus*) demonstrated that the piRNA pathway is essential for both male and female fertility[23–25]. Disruption of *Piwil1* and *Mov10l1* results in sterility in both male and female golden hamsters with severe defects in spermatogenesis or embryonic arrest at the two-cell stage, while *Piwil3* deficiency causes subfertility in females. Thus, the similarity in *Piwi* gene expression

[1]State Key Laboratory of Molecular Biology, Shanghai Key Laboratory of Molecular Andrology, Center for Excellence in Molecular Cell Science, Shanghai Institute of Biochemistry and Cell Biology, Chinese Academy of Sciences, University of Chinese Academy of Sciences, Shanghai 200031, China. [2]State Key Laboratory of Reproductive Medicine and Offspring Health, Jiangsu Laboratory Animal Center, Jiangsu Animal Experimental Center of Medicine and Pharmacy, Department of Cell Biology, Animal Core facility, Key Laboratory of Model Animal, Collaborative Innovation Center for Cardiovascular Disease Translational Medicine, Nanjing Medical University, Nanjing 211166, China. [3]These authors contributed equally: Xiaolong Lv, Wen Xiao, Yana Lai, Zhaozhen Zhang, Hongdao Zhang. ✉e-mail: sam@njmu.edu.cn; jianminlilab@njmu.edu.cn; lgwu@sibcb.ac.cn

patterns and small RNA composition among hamster, bovine, monkey, and human oocytes suggest that golden hamsters can serve as a more accurate model for elucidating piRNA function and mechanisms in female mammals.

However, several fundamental lines of evidence critical for understanding the role of PIWI and piRNAs remain incomplete in the golden hamster model. In particular, a comprehensive spatio-temporal landscape of the expression patterns for all four PIWIs is still lacking, and the dynamic changes in their associated piRNAs are largely uncharacterized. In addition, although it is well-established that *Piwil2* and *Piwil4* are expressed in the testes and ovaries of most mammals, including humans[9,26–28], but whether *Piwil2* and *Piwil4* are required for fertility in golden hamsters remains unknown, as do their exact roles in gametogenesis and early embryogenesis. In addition, *Mov10l1*[−/−] and *Piwil1*[−/−] golden hamsters display greater defects in spermatogenesis than mice harboring these mutations[16,23–25,29,30], suggesting that golden hamsters may provide additional mechanistic insights into the role of piRNAs in male reproduction. Thus, uncovering the potential functions of all individual *Piwi* genes in golden hamsters can be informative towards understanding their interplay in regulating mammalian germline or embryonic development.

In this study, we establish comprehensive spatio-temporal expression profiles for all four PIWI proteins and their associated piRNAs in both male and female golden hamsters. Moreover, we generated *Piwil2*[−/−], *Piwil3*[−/−], and *Piwil4*[−/−] golden hamsters to compare the potential reproductive defects associated with each of the four *Piwi*-mutants and characterize the sex-specific functions and interplay among the four PIWI proteins. Our results highlight the intertwined but non-redundant functions of PIWI proteins in gametogenesis and embryogenesis in golden hamsters. These findings expand our mechanistic understanding of PIWI functions and piRNA dynamics in mammals.

## Result

### Coordinated expression of PIWI proteins in adult spermatogenesis

Since the temporal expression and subcellular localization of PIWI proteins are both critical for their function, we generated antibodies to observe individual PIWI proteins in golden hamsters and validated their specificity (Supplementary Fig. 1a). In postnatal testes of golden hamsters (at 3 days post-partum, 3 d.p.p.), only PIWIL2 and PIWIL4 were detectable (Fig. 1a). While PIWIL4 was present in both the nucleus and cytoplasm, PIWIL2 displayed exclusively cytoplasmic localization. Super-resolution microscopy further revealed that PIWIL2 colocalizes with DCP1A, TDRD1, and ATP5A in prospermatogonia, thus providing evidence of its presence in piP bodies and IMC (Supplementary Fig. 1b, 1e, and 1f)[31,32]. And PIWIL4 colocalize with DCP1A and DDX6 in prospermatogonia, supporting its presence in piP bodies (Supplementary Fig. 1c and 1d).

In spermatocytes of adult testes, PIWIL1 and PIWIL2 were highly expressed and only present in the cytoplasm (Fig. 1b). During leptotene/zygotene in early spermatocytes, only PIWIL2 was expressed and observable in germinal granules, which appeared as irregular clouds (Supplementary Fig. 1g). Then, in pachytene spermatocytes, PIWIL1 expression was detectable, and subsequently co-localizing with PIWIL2 to form enriched germinal granules containing condensed bodies (Fig. 1b and Supplementary Fig. 1g). In late pachytene and diplotene stages, those germinal granules dissociated from each other and diffused in the cytoplasm concomitant with the appearance of CB precursors in the perinuclear region. When entering meiosis II, PIWI-enriched germ granules were undetectable, and then early chromatoid bodies (CBs) appeared as multiple perinuclear vesicles with PIWIL1 localized in the interior and PIWIL2 located on the exterior in early round sperms. Finally, in late round sperms, CBs aggregated into a single vesicle in which only PIWIL1 was present

(Supplementary Fig. 1g). This morphological change in germinal granules occurred simultaneously with changes in PIWI protein expression (Fig. 1g), suggesting that PIWIL1 and PIWIL2 perform independent but coordinated functions in adult spermatogenesis.

### The expression and sub-localization of PIWIL1 and PIWIL3 are well organized during oogenesis and early embryogenesis

We next examined PIWI expression and localization in female germline cells and embryos for comparison with that observed in males. Immunostaining with PIWI antibodies in adult ovaries of golden hamsters revealed that PIWIL1 and PIWIL3 accumulated in the cytoplasm of both quiescent and growing oocytes (Fig. 1c), which aligned well with previous reports[23–25]. Unexpectedly, PIWIL2 was exclusively expressed in the cytoplasm of oocytes in the primordial follicle and early primary follicle stages. It is noteworthy that PIWIL4 was conspicuously absent throughout the entirety of oocyte development.

To further characterize the stage-specific subcellular localization of PIWI proteins, we performed co-immunostaining with antibodies for PIWIs, DDX4 (expressed in germ granules), and TDRKH (presented on the outer membrane of mitochondria) in ovary tissue of adult hamsters. In oocytes at the primordial follicle and early primary follicle stages, PIWIL1 and PIWIL2 co-localized with DDX4, but not TDRKH, in the germinal granules of the cytoplasm, forming a perinuclear ring structure with characteristic IMC morphology (Supplementary Fig. 2a and 2b). In contrast, PIWIL3 exclusively co-localized with TDRKH on the outer mitochondrial membrane (Supplementary Fig. 2b and 2e). In growing oocytes, PIWIL2 was gradually depleted as PIWIL1 and PIWIL3 aggregate into prevalent in the germinal granules, co-localizing with DDX4 and TDRKH to collectively form Balbiani body (BB) subcellular structures (Supplementary Fig. 2c and 2d), which are conserved in mammalian oocytes[33,34].

In MII oocytes and two-cell embryos, PIWIL1 and PIWIL3 were consistently expressed, whereas the signals for PIWIL2 and PIWIL4 were undetectable (Fig. 1d). PIWIL1 was evenly distributed in the MII oocytes as a prominent granule (Fig. 1d), resembling the distribution of Aub in early embryos in Drosophila[35]. RNase treatment showed that the cytoplasmic foci were RNA granules (Supplementary Fig. 2f), indicating that these were likely sites of RNA accumulation and regulation. Unexpectedly, PIWIL1 mainly accumulated in the nucleolus of two-cell embryos (Fig. 1e), which has not been observed in other animal models and suggests a dramatic shift in PIWIL1 function in the transition from oogenesis to embryogenesis. In addition, PIWIL3 showed obvious co-localization with mitochondria in MII oocytes and formed cytoplasmic foci in two-cell embryos (Fig. 1d, e). In four-cell embryos, PIWIL3 was the only PIWI expressed at high levels, and again co-localized with mitochondria (Fig. 1f, Supplementary Fig. 2g and Supplementary Fig. 2h).

### Generation of *Piwi*-deficient golden hamsters

Previously, we generated *Piwil1*[−/−] golden hamsters, which resulted in complete sterility in both males and females[24]. To systematically investigate the roles of individual *Piwi* genes in both male and female reproduction, we further generated *Piwil2*[−/−], *Piwil3*[−/−], and *Piwil4*[−/−] golden hamsters by two-cell embryo injection of CRISPR−Cas9 mRNAs with sgRNAs targeting exon 13 of *Piwil2*, exon 6 of *Piwil3*, and exon 4 of *Piwil4*, respectively (Supplementary Fig. 3a and 3b). The F1 heterozygous mutant hamsters were crossed to generate homozygous *Piwi*-deficient golden hamsters. The CRISPR-Cas9-induced frameshift mutation introduced premature stop codons in each PIWI protein, including a 35-nt deletion in exon 13 of *Piwil2*[−/−] hamsters, a 172-nt deletion in exon 6 of *Piwil3*[−/−] animals, and a 70-nt deletion in exon 4 of *Piwil4*[−/−] golden hamsters (Supplementary Fig. 3b). The disruption of *Piwi* gene transcription was confirmed by RNA-seq (Supplementary Fig. 3c−f, Supplementary Data 2-4), and the corresponding loss of PIWI

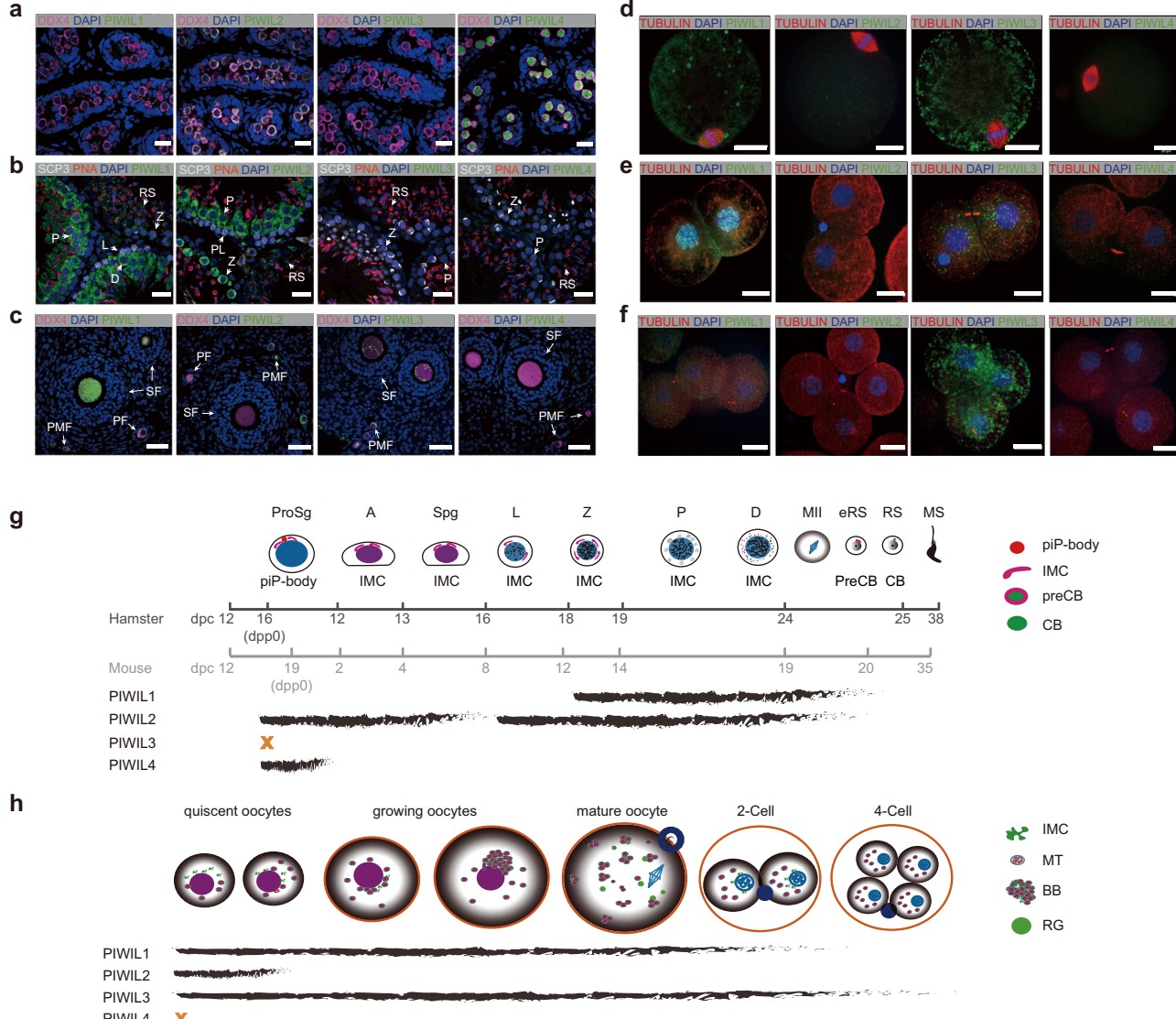

**Fig. 1 | Expression of PIWI proteins during gametogenesis and early embryogenesis. a**, **b** Immunofluorescence staining to examine the expression of PIWI proteins in WT postnatal (**a**) or adult testes (**b**). Germ cells in postnatal testes were marked by DDX4. Adult testes were stained with anti-SCP3 antibodies and PNA. Nuclei were stained with 4, 6-diamidino-2-phenylindole (DAPI). PL preleptotene, L leptotene, Z zygotene, P pachytene, D diplotene, RS round sperm. Scale bars, 20 μm. **c** Immunofluorescence staining to examine the expression of PIWI proteins in WT adult ovaries. Germ cells were marked by DDX4, and nuclei were stained with DAPI. PMF primordial follicle, PF primary follicle, SF secondary follicle. Scale bars, 40 μm. **d**–**f** Immunofluorescence staining to examine the expression of PIWI

proteins in WT MII oocytes (**d**), two-cell embryos (**e**), or four-cell embryos (**f**). Scale bars, 20 μm. **g**, **h** Schematic diagrams showing the expression of PIWI proteins during male gametogenesis (**g**) or female gametogenesis and early embryogenesis (**h**). Charcoal lines indicate the presence of individual PIWI proteins, while orange crosses indicate their absence. d.p.c. days postcoitus, ProSg prospermatogonia, A type A undifferentiating spermatogonia, Spg differentiating spermatogonia, eRS early round sperm, MS mature sperm, IMC inter mitochondrial cement, preCB prechromatoid body, CB chromatoid body, MT mitochondrial, BB Balbina body, RG RNA granule.

protein accumulation was verified by Western Blot (WB) and immunostaining (Fig. 2a, b).

All *Piwi*-deficient golden hamsters appeared normal without any obvious, discernible morphological or behavioral abnormalities. However, *Piwi2*⁻/⁻ and *Piwi4*⁻/⁻ males were completely sterile, although female mutants had similar fertility to that of WT control animals (Fig. 2c, d). By contrast, *Piwi3*⁻/⁻ females displayed reduced fertility with significantly fewer offspring, consistent with previous reports[23,25], although the mutation site in *Piwi3* was different. These reproductive phenotypes aligned well with our observations of high PIWIL1, PIWIL2, and PIWIL4 expression in male gametes, but only high expression of PIWIL1 and PIWIL3 in female gametes (Fig. 1g, h). This cumulative evidence supported the likelihood of PIWI proteins are responsible for sex-specific functions in gametogenesis.

## PIWIL1 29-nt piRNAs partially compensate for the loss of PIWIL3 19-nt piRNAs in oocytes

Since PIWIL1 is universally expressed in male and female gametes, but PIWIL3 is exclusive to females, we investigated their specific roles in oocytes and embryos. Histological examination showed no visible abnormalities in the ovaries of any of the *Piwi*-deficient golden hamsters (Supplementary Fig. 4a). Consistent with previous studies[23,24], maternal *Piwil1*⁻/⁻ embryos were arrested at the two-cell stage. While most maternal *Piwil3*⁻/⁻ embryos were also arrested at the two-cell stage, a few embryos could proceed to the four-cell or later stages (Supplementary Fig. 4b). Although pregnancy rate and litter size were significantly decreased in the absence of *Piwil3*, abortive embryos were observed in ~35% of individuals at the second estrous cycle after successful fertilization (Supplementary Fig. 4c and 4d), indicating

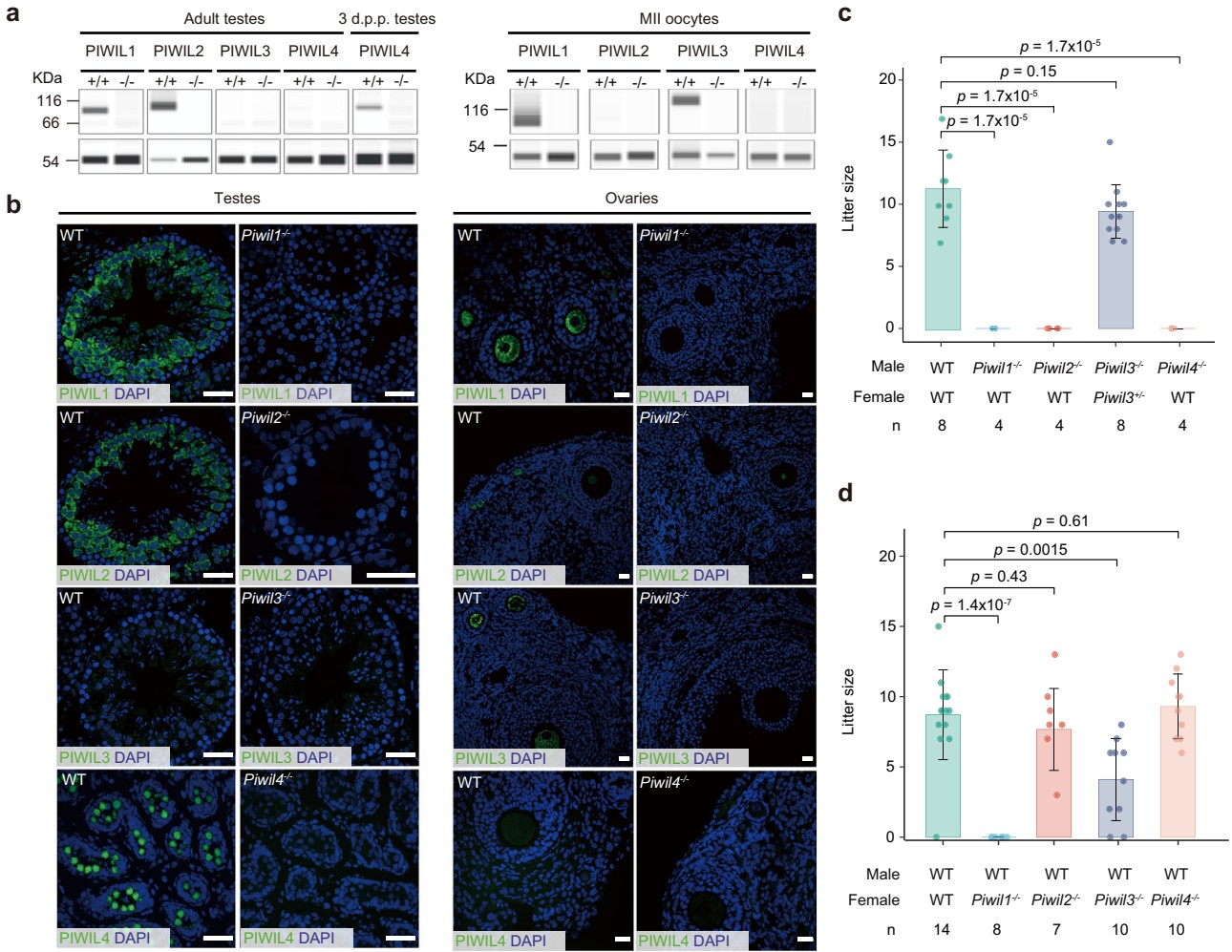

**Fig. 2 | Generation and fecundity analysis of *Piwil1*⁻/⁻, *Piwil2*⁻/⁻, *Piwil3*⁻/⁻, and *Piwil4*⁻/⁻ golden hamsters. a** Western blots showing the expression of PIWI proteins in testes and MII oocytes of *Piwil1*⁻/⁻, *Piwil2*⁻/⁻, *Piwil3*⁻/⁻, and *Piwil4*⁻/⁻ golden hamsters, respectively. α-TUBULIN (left panel) or β-ACTIN (right panel) was used as the loading control. **b** Immunostaining analysis exhibiting the loss of PIWIL1-4 expression in the mutant ovaries and testes. The faint, diffuse signal of PIWIL2 in nuclei of secondary follicle stage mutant oocytes is likely background autofluorescence. Scale bars, 40 µm. **c, d** The fecundity of male (**c**) and female (**d**) *Piwi*-deficient golden hamsters, respectively. Data are mean ± s.e.m. All *p* values are from two-tailed Student's *t*-tests. *n* represents the number of mating. Source data are provided as a Source Data file.

that, despite their apparent WT morphology, many embryos could not successfully develop into pups.

*Piwil1*⁻/⁻ results in the loss of 23- and 29-nt piRNAs and a significant increase in TE transcript abundance, while *Piwil3*⁻/⁻ results in the elimination of 19-nt piRNAs but does not obviously affect TE expression (Fig. 3a, Supplementary Fig. 5a and Supplementary Fig. 5b, Supplementary Data 5 and 6), despite the similar genomic origins of PIWIL1 and PIWIL3 piRNAs[24]. In previous work, we found that the production of some PIWIL3 piRNAs partially depended on PIWIL1, possibly through heterotypic Ping-Pong between PIWIL1 and PIWIL3 proteins. Indeed, some ERV-derived PIWIL3 piRNAs significantly decreased in *Piwil1*⁻/⁻ oocytes (Fig. 3b, d, and Supplementary Fig. 5d). In contrast, the relative abundance of PIWIL1 29-nt obviously increased in *Piwil3*⁻/⁻ oocytes (Fig. 3a and Supplementary Fig. 5b). Next, PIWIL1 protein expression levels in *Piwil3* mutant oocytes with that of PIWIL3 levels in *Piwil1* mutant oocytes were compared (Supplementary Fig. 4e and 4f). The results show that PIWIL3 protein levels are significantly decreased in the absence of *Piwil1*, while PIWIL1 expression shows a non-significant, increasing trend in *Piwil3* mutant oocytes. More than 70% of PIWIL1 and PIWIL3 piRNAs exhibit identical 5' ends and are likely to compete for a common set of pre-piRNAs[24]. Thus, in the absence of PIWIL3, more pre-piRNAs are loaded in the abundant PIWIL1 and

trimmed to 29-nt instead of 19-nt. By contrast in *Piwil1* mutant oocytes, impaired PIWIL3 expression results in lower efficiency processing of the larger pool of pre-piRNAs, and consequently leading to almost unchanged expression levels of 19-nt piRNAs (Fig. 3a).

Among the increased 29-nt piRNAs in *Piwil3*⁻/⁻ oocytes, TE-derived piRNAs were mainly related to active ERVK or ERV1 (Fig. 3b), with a concomitant significant increase in their Ping-Pong signal (Fig. 3c and Supplementary Fig. 5c). We further conducted smRNA-seq and mRNA-seq (Supplementary Data 2) in oocytes at the germinal vesicle (GV) and metaphase II (MII) stage and embryos at 11 h.p.e.a. (the one-cell stage in WT), 34 h.p.e.a. (the two-cell stage in WT), and 54 h.p.e.a. (the four-cell stage in WT), and found that the increased production of PIWIL1 29-nt piRNAs lasted in maternal *Piwil3*⁻/⁻ embryos until 34 h.p.e.a. (Fig. 4a). Aligned well with this, some essential piRNA processing genes, including *Gtsf1*, *Ddx4*, *Tdrd1*, *Henmt1*, *Pld6*, *Mael*, were moderately expressed in maternal *Piwil3*⁻/⁻ but not in WT embryos at 34 h.p.e.a., whereas their expression was similar at earlier stages (Supplementary Fig. 5e, Supplementary Data 7). These results suggested that 29-nt piRNAs can partially complement the role of 19-nt piRNAs, at least silencing a subset of active ERVs, to try to ensure the proper development of *Piwil3*⁻/⁻ oocytes or maternal *Piwil3*⁻/⁻ embryos.

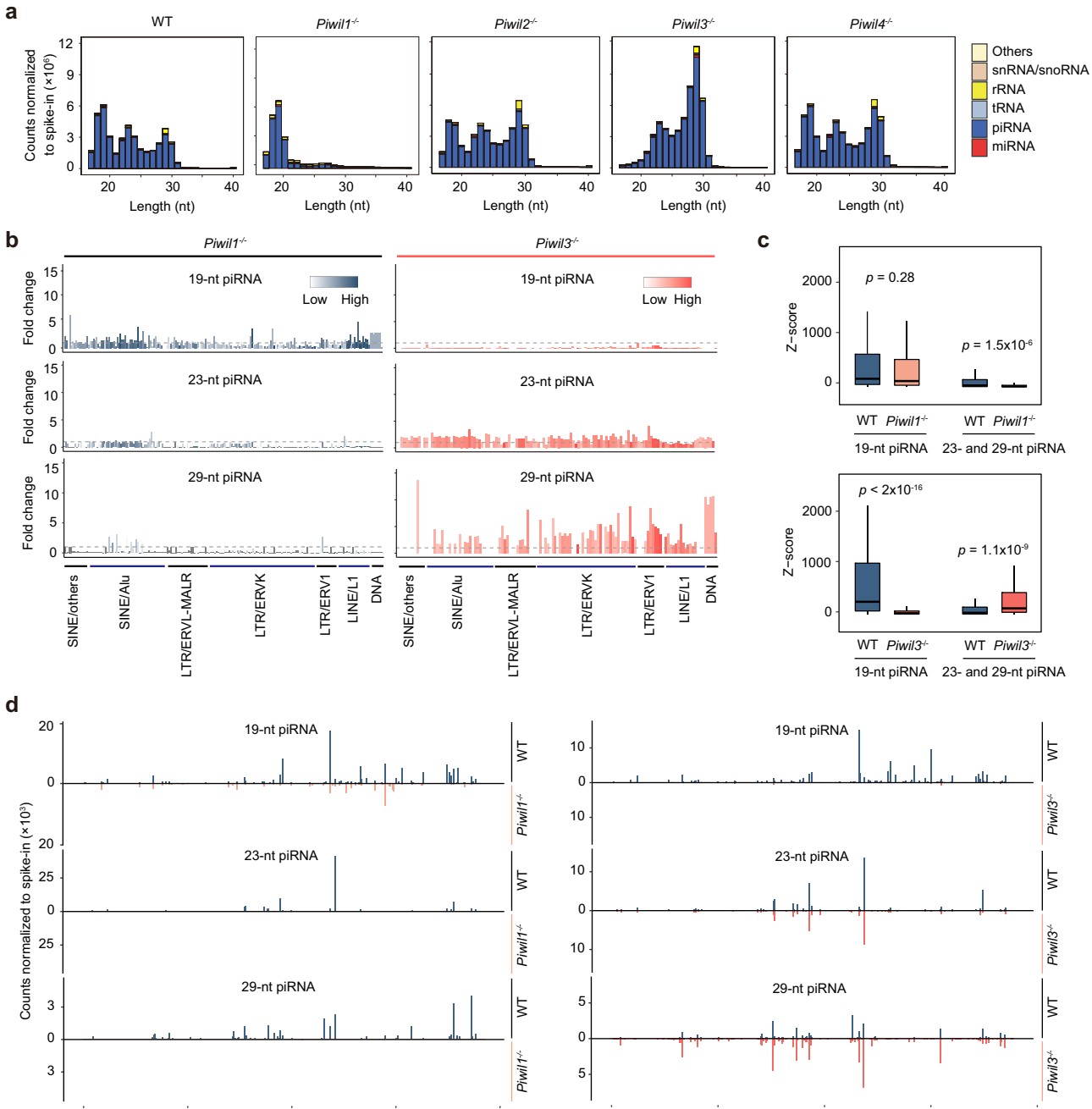

**Fig. 3 | PIWIL1-piRNA compensating for repressing TEs in *Piwil3*⁻/⁻ oocytes.**
**a** The composition of small RNA categories in WT, *Piwil1*⁻/⁻, *Piwil2*⁻/⁻, *Piwil3*⁻/⁻, or *Piwil4*⁻/⁻ MII oocytes. Small RNA counts were normalized to the exogenous spike-in. Data are the average values of biological replicates: 3 for WT, 2 for *Piwil1*⁻/⁻, 3 for *Piwil2*⁻/⁻, 4 for *Piwil3*⁻/⁻, and 3 for *Piwil4*⁻/⁻. **b** The fold-change of piRNAs expression level mapped to each TE family in *Piwil1*⁻/⁻ or *Piwil3*⁻/⁻ compared to WT MII oocytes. 19-nt piRNAs (18-20nt), 23-nt piRNAs (22-24nt), and 29-nt piRNAs (28-30nt) derived from potentially active TE families were designated and plotted. The fold-change of piRNA expression level in *Piwil1*⁻/⁻ or *Piwil3*⁻/⁻ oocytes is indicated by the degree of blue or red, respectively. The gray dotted line indicates a fold-change of 1. The potentially active TEs were defined as nucleotide replacement rate <0.15 and Z-score ≥1.96. **c** The ping-pong signature of 19-nt or 23-nt and 29-nt piRNAs derived from potentially active TEs in WT, *Piwil1*⁻/⁻, and *Piwil3*⁻/⁻ MII oocytes. The level of the ping-pong signature is represented by a Z-score of 10-nt overlapped piRNAs from

opposite strands; piRNAs with overlaps of different lengths serve as the background. Z > 1.96 corresponds to $p < 0.05$. For the box plots, the centre line represents the median value, the box borders represent the upper and lower quartiles (25th and 75th percentiles, respectively), and the ends of the top and bottom whiskers represent the maximum and minimum scores, respectively. Two-sided Wilcoxon test was employed to examine the statistical significance, and no adjustments were made for multiple comparisons. **d** Example (ltr-1_family-22 | LTR/ERVK) of 19-nt, 23-nt, and 29-nt piRNAs distribution derived from ERVK families in WT, *Piwil1*⁻/⁻ and *Piwil3*⁻/⁻ MII oocytes. The horizontal axis represents the position of the consensus TE, and the vertical axis represents the normalized number of all mapped piRNAs with the same 5' end at each position. Data in (**b**–**d**) are the average values from three WT and two *Piwil1*⁻/⁻ biological replicates or four WT and four *Piwil3*⁻/⁻ biological replicates.

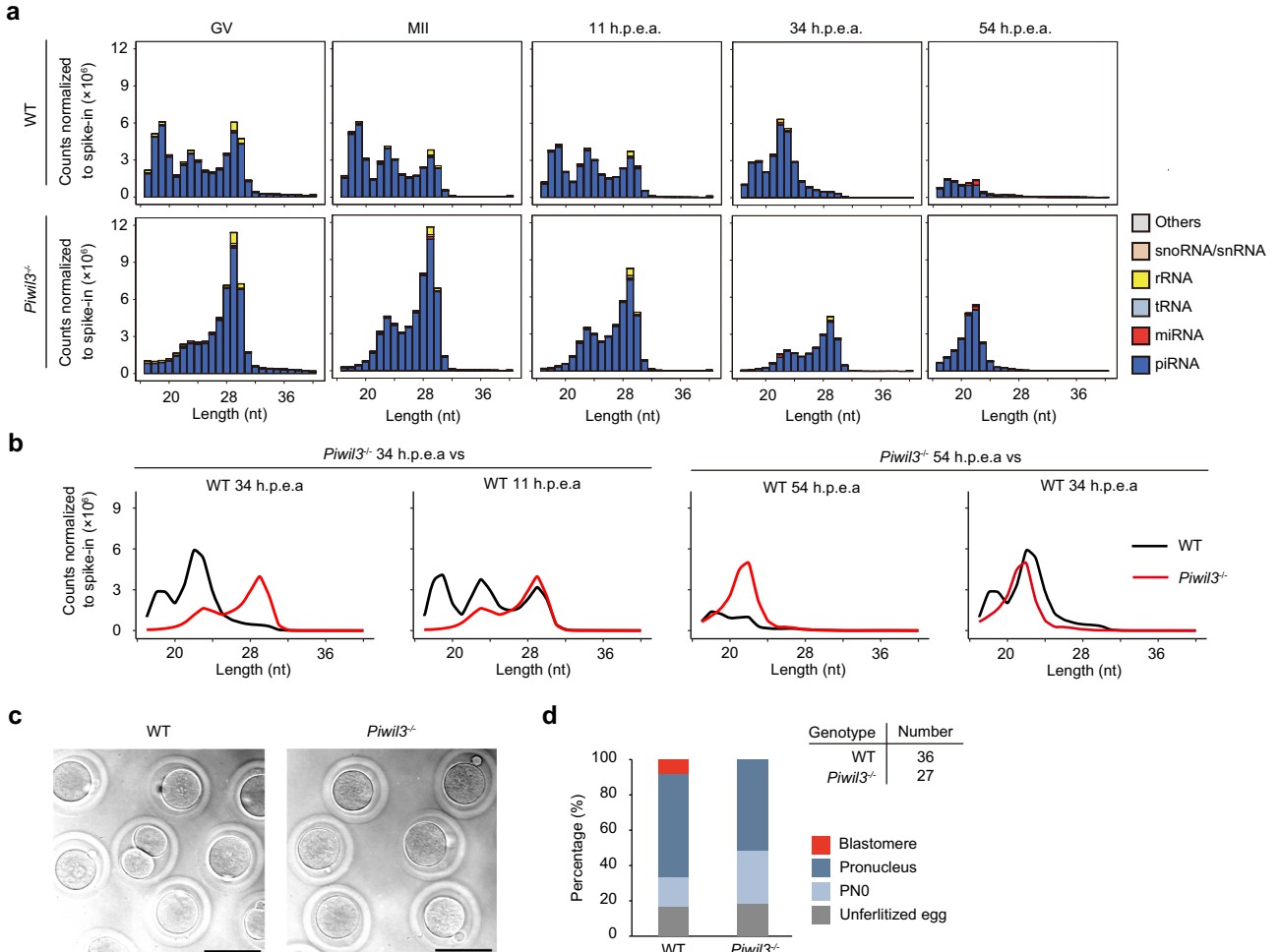

**Fig. 4 | Development of maternal *Piwil3*⁻/⁻ embryos is delayed. a** The composition of small RNA categories according to their length distribution during oocyte and embryo development in WT or *Piwil3*⁻/⁻ golden hamsters. Early-stage oocytes were collected from ovaries, and MII oocytes were collected by spontaneous ovulation. The embryos were collected from the oviducts of WT and *Piwil3*⁻/⁻ female hamsters mated with WT male hamsters. The small RNA counts were normalized to the exogenous spike-in. **b** Line charts showing the expression pattern of piRNAs in *Piwil3*⁻/⁻ embryos compared to WT. Data in (**a**–**b**) are the average values of the biological replicates at each time point: $n = 3$ (GV), $n = 4$ (MII), $n = 5$ (11 h.p.e.a.), $n = 4$ (34 h.p.e.a.), or $n = 2$ (54 h.p.e.a.) for WT; $n = 3$ (GV), $n = 4$ (MII), $n = 5$ (11 h.p.e.a.), $n = 3$ (34 h.p.e.a.), or $n = 2$ (54 h.p.e.a.) for *Piwil3* mutants. *Piwil3*⁻/⁻, *Piwil3*⁻/⁻ oocytes or maternal *Piwil3*⁻/⁻ embryos. **c** Representative images showing the delayed development of maternal *Piwil3*⁻/⁻ embryos at 18 h.p.e.a. Scale bar, 100 μm. **d** Histogram showing embryogenesis ratio of WT and maternal *Piwil3*⁻/⁻ embryos 18 h.p.e.a. PN0, shortly after the fertilization but before entering the pronuclear stage. Source data are provided as a Source Data file.

## Maternal disruption of *Piwil3* results in delayed development of embryos

Despite the apparent compensatory effects of PIWIL1, the attenuated fertility in *Piwil3*⁻/⁻ females suggested PIWIL1 could not fully complement PIWIL3 function. To explore this, the expression patterns of piRNAs and genes in oocytes at the GV and MII stage and embryos at 11 h.p.e.a., 34 h.p.e.a., and 54 h.p.e.a. were analyzed. In WT embryos, production of 29-nt piRNAs ceases after 34 h.p.e.a., but ends after 54 h.p.e.a. in maternal *Piwil3*⁻/⁻ embryos (Fig. 4a). Additionally, production of 23-nt piRNAs substantially decreases after 54 h.p.e.a. in WT embryos, but are still expressed at high levels comparable to WT embryos at 34 h.p.e.a. at this stage in maternal *Piwil3*⁻/⁻ embryos (Fig. 4b). These findings suggest a developmental delay in maternal *Piwil3*⁻/⁻ embryos compared to WT embryos. This developmental lagging was further confirmed by the hierarchical clustering results of the top 10,000 individual piRNAs that the expression change of most piRNAs was delayed in maternal *Piwil3*⁻/⁻ embryos (Supplementary Fig. 6a, Supplementary Data 8). The expression of piRNAs in cluster 4 (51.69%) was remarkably decreased from 11 h.p.e.a. in WT embryos, while in maternal *Piwil3*⁻/⁻ embryos this decrease was observed till 34 h.p.e.a. In addition, the expression of

piRNAs in cluster 3 (24.77%) started to increase from 11 h.p.e.a. in WT embryos, again in maternal *Piwil3*⁻/⁻ embryos this expression change was lagged.

Next, we compared gene and TE expression in *Piwil3*⁻/⁻ oocytes or maternal *Piwil3*⁻/⁻ embryos with WT. In oocytes at GV and MII stages and embryos at 11 h.p.e.a., no significant difference in gene expression was observed (Supplementary Fig. 6b), consistent with a previous study[23]. In WT embryos, the expression of most genes increased from 11 h.p.e.a., whereas this increase was observed from 34 h.p.e.a. in maternal *Piwil3*⁻/⁻ embryos (Supplementary Data 9). In contrast, TEs were silenced to a level in *Piwil3*⁻/⁻ oocytes or maternal *Piwil3*⁻/⁻ embryos comparable to WT at a time consistent with expression of most TEs (Supplementary Fig. 6c, Supplementary Data 10), further confirming PIWIL1 can sufficiently compensate PIWIL3 in repressing TEs but not in regulating gene expression. Collectively, the development of maternal *Piwil3*⁻/⁻ embryos appeared to begin lagging at 11 h.p.e.a., which was further verified by the delayed in vivo development of maternal *Piwil3*⁻/⁻ embryos at 18 h.p.e.a. (Fig. 4c, d). These cumulative findings suggested that aside from silencing TEs, PIWIL3 has non-redundant functions in female golden hamsters required for proper embryonic development.

## Defects in spermatogenesis in *Piwi*-deficient golden hamsters

In light of our above findings of the effects of *Piwi* knockout in oocytes and embryos, we next examined the effects of each *Piwi* in germline cells of adult and neonatal testes. In males, *Piwil3*⁻/⁻ golden hamsters displayed normal morphology of the testis and cauda epididymis, whereas the testes and cauda were obviously smaller than WT in *Piwil1*⁻/⁻, *Piwil2*⁻/⁻ and *Piwil4*⁻/⁻ males (Supplementary Fig. 7a and 7b). Moreover, *Piwil2*⁻/⁻ and *Piwil4*⁻/⁻ testes were much smaller than those of *Piwil1*⁻/⁻ animals, indicating that deficiency of *Piwil2* and *Piwil4* resulted in markedly more severe developmental impairment. In agreement with its proposed female-specific role, periodic acid–Schiff (PAS) staining of testicular and epididymal sections from *Piwil3*⁻/⁻ adult male hamsters showed that sperms and spermatocytes in the cauda epididymis or seminiferous tubules also appeared normal (Fig. 5a). By contrast, in *Piwil1*⁻/⁻ males, no mature sperms were observed in the cauda epididymis, and few, if any, round spermatids were found in the seminiferous tubules. In addition to the dysregulated production and subcellular distribution of glycoproteins in pachytene-like spermatocytes, we also found that DDX4 and PIWIL2 exhibit abnormal nuclear localization and form aberrant condensed foci in the absence of PIWIL1, and in late meiotic spermatocytes, CB-like germinal granules and PIWIL2 show abnormal colocalization (Supplementary Fig. 7c–e). Those findings suggested *Piwil1*⁻/⁻ resulted in severe developmental disorders in spermatogenesis.

In addition, sperm production was also abolished in the cauda epididymis of males deficient for *Piwil2* or *Piwil4*, while the seminiferous tubules were smaller with clearly altered architecture, and had notably fewer germ cells, which was further verified by immunostaining for germ cell marker DDX4 in testes (Fig. 5a and Supplementary Fig. 7c). *Piwil2*⁻/⁻ seminiferous tubules contained only sertoli cells and a small number of pre-meiotic spermatocytes could be found in less than 4% of the tubules, a phenotype similar to that of *Mov10l1*⁻/⁻ golden hamster, but different from *Piwil2*⁻/⁻ mice. Immunostaining of testes with LIN28A-specific antibody revealed that undifferentiated spermatogonia were largely absent in *Piwil2*⁻/⁻ testes, supporting a role of PIWIL2 in the self-renewal of those stem cells (Supplementary Fig. 7c). By contrast, although *Piwil4*⁻/⁻ also resulted in a substantial loss of germ cells, these hamsters had more germ cells than *Piwil2*⁻/⁻ and a similar number of undifferentiated spermatogonia in testes to that of WT (Fig. 5a and Supplementary Fig. 7c).

To better understand the loss of germ cells in *Piwi* mutant testes, we examined the testicular cell phenotypes at 14, 21, and 40 d.p.p. in neonatal males, corresponding to the first wave of spermatogenesis. Differentiated spermatogonia appear in newborn testes by 14 d.p.p. and spermatocytes can reach the pachytene stage of spermatogenesis by 21 d.p.p. At 40 d.p.p., mature sperms are produced in the testes[36]. No apparent abnormalities in spermatogenic cell development were observed at 14 d.p.p. in any of the *Piwi* deficiency hamsters (Fig. 5b), suggesting that the impairment of spermatogenesis begins after meiosis. At 21 d.p.p., gametes had abnormally concentrated nuclei in *Piwil1*⁻/⁻ seminiferous tubules, a phenotype indicating germ cell death before the pachytene stage, consistent with findings in adult testes. In *Piwil2*⁻/⁻ and *Piwil4*⁻/⁻ hamsters, spermatocyte counts were strikingly decreased in seminiferous tubules compared to WT, and to a greater extent in *Piwil2*⁻/⁻ hamsters. At 40 d.p.p., when the first round of meiosis is completed, no mature sperms could be found in either *Piwil1*⁻/⁻, *Piwil2*⁻/⁻, or *Piwil4*⁻/⁻ testes, a phenotype identical to that of adult mutant testes. Altogether, those findings suggest that *Piwil1*⁻/⁻, *Piwil2*⁻/⁻, or *Piwil4*⁻/⁻ all resulted in severe spermatogenesis defects, whereas it was distinguishable.

## *Piwil1*⁻/⁻, *Piwil2*⁻/⁻, or *Piwil4*⁻/⁻ results in the development arrest or cell death of spermatogenic cells

To determine the composition of spermatogenic cells in *Piwi*-deficient testes, FACS was used to isolate germ cells from adult testes of the WT and deficiency lines (Supplementary Fig. 8a–d)[37]. *Piwil1*⁻/⁻ testes were enriched for spermatocytes arrested at the pachytene or diplotene stages, as reported previously (Fig. 5c)[24]. In *Piwil2*⁻/⁻ and *Piwil4*⁻/⁻ testes, most survived spermatocytes were arrested at the zygotene stage, although approximately 2–5% could still develop to the pachytene and diplotene stages (Fig. 5c, Supplementary Fig. 8c, and Supplementary Fig. 8d). These results were further confirmed by immunostaining for the prophase meiotic spermatocyte marker proteins SCP3 and rH2AX in testes tissue (Supplementary Fig. 8e). Unexpectedly, a small population of round sperms was observed in the seminiferous tubules of *Piwil4*⁻/⁻ males.

A previous study showed that *Piwil1*⁻/⁻ resulted in cell death of spermatocytes[24]. To determine the cell viability of these spermatocytes in each *Piwi* mutant testes, Hoechst33342 and propidium iodide (PI) staining of cells in seminiferous tubules of *Piwi*-deficient hamsters in conjunction with FACS-based quantification were performed (Supplementary Fig. 9a–e). In *Piwil1*⁻/⁻ testes, as many as 26% of the total spermatocytes were PI-positive, most of which were in the zygotene or pachytene stages (Supplementary Fig. 9b). Thus, under *Piwil1* deficiency, a substantial proportion of spermatocytes die before reaching the pachytene stage, which was further verified by TUNEL assays (Supplementary Fig. 9f). Although a significant proportion can survive to reach the pachytene or diplotene stage, these spermatocytes were abnormal and few progressed to the next stage; instead, they eventually detached from the tubules, flowed into the caput epididymis, and underwent cell death (Supplementary Fig. 9g). In *Piwil2*⁻/⁻ and *Piwil4*⁻/⁻ testes, the death of spermatocytes was also observed, although with a much lower proportion since most germ cells had died in postnatal testes (Supplementary Fig. 9c, 9e, and 9f). These cumulative findings suggested the development of spermatogenic cells in *Piwi*-deficient golden hamsters was severely impaired, even greater than in mice (Fig. 5d).

## PIWIs play interconnected and non-redundant roles during spermatogenesis

To gain insight into the molecular defects leading to the observed abnormalities, we conducted smRNA-seq and RNA-seq in the testes of WT and *Piwi*-deficient golden hamsters (Supplementary Data 2). Firstly, in order to understand the detailed information of PIWI bound piRNA in male germline cells, we performed immunoprecipitation with anti-PIWIL1/2/4 antibodies using WT testes. The result indicated that in 3 d.p.p. testes, PIWIL2 and PIWIL4 were predominantly associated with 27-nt and 28-nt piRNAs, respectively (Supplementary Fig. 10a). However, in adult testes, PIWIL1 and PIWIL2 were most frequently bound to 29-nt and 27-nt piRNAs, respectively (Supplementary Fig. 10b) (hereafter referred to as PIWIL1 29-nt piRNAs; PIWIL2 27-nt piRNAs, postnatal or adult; and PIWIL4 28-nt piRNAs). Among them, PIWIL1 29-nt and PIWIL2 27-nt piRNAs preferentially carried a 5′ terminal uridine (1U), which was less common in PIWIL4 28-nt piRNAs were instead enriched with adenine at nucleotide 10 (10 A) (Supplementary Fig. 10c). Notably, in hamster postnatal testes, piRNAs originated from diverse genomic loci and were mainly derived from TEs, especially for the PIWIL4 28-nt piRNAs, with the top 100 most highly expressed piRNA clusters notably generating only 11–14% of all piRNAs (Supplementary Fig. 10d and 10e). However, in adult testes, piRNAs were derived from a relatively select group of extended genomic regions, with the top 100 most highly expressed piRNA clusters producing >94% piRNAs, with sequences primarily derived from intergenic regions. Even though the length of PIWIL2-bound piRNAs was identical between postnatal and adult testes, the two piRNA species were produced from different piRNA clusters and their sequences showed almost no overlap.

Next, we comprehensively analyzed the sequencing data generated by using *Piwi*-deficient testes. In *Piwil2*⁻/⁻ 3 d.p.p. testes, all piRNA populations were absent, as were PIWIL4 28-nt piRNAs (Fig. 6a and

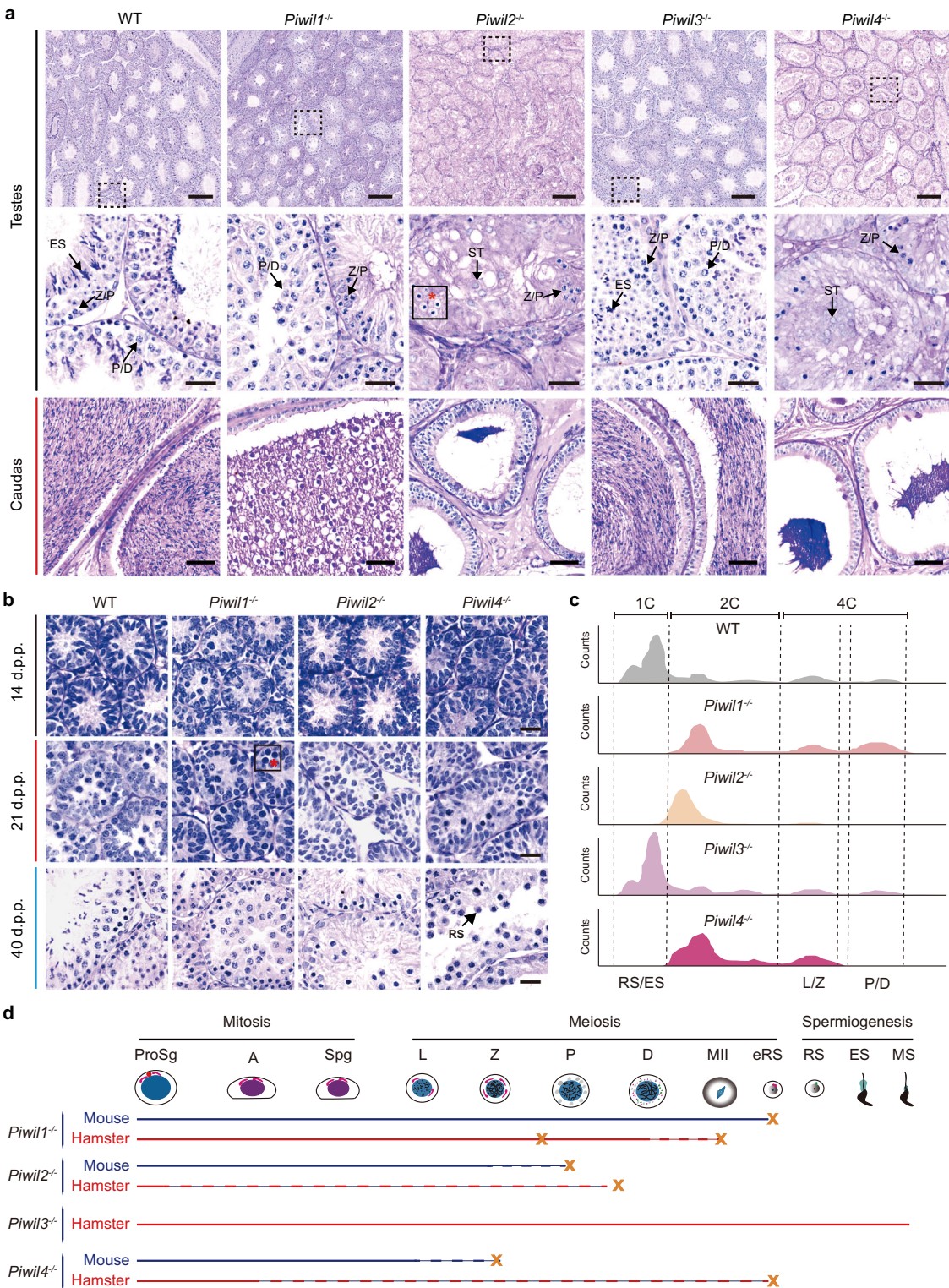

**Fig. 5 | Spermatogenesis defects in *Piwil1*⁻/⁻ *Piwil2*⁻/⁻, and *Piwil4*⁻/⁻ golden hamsters. a** Periodic acid–Schiff (PAS) staining of adult testes. Spermatocytes with condensed nuclei observed in *Piwil2*⁻/⁻ testes are indicated by the red asterisk. Z/P, zygotene or pachytene; P/D, pachytene or diplotene; ES, elongating sperm; ST, Sertoli cell. Scale bar, 200 μm (top), 25 μm (middle and bottom). **b** PAS staining of postnatal testes at 14 d.p.p., 21 d.p.p., and 40 d.p.p. Spermatocytes with abnormal nuclei observed in *Piwil1*⁻/⁻ testes are indicated by the red asterisk. RS, round sperm.

Scale bar, 20 μm. **c** Flow cytometric analysis of adult testicular cells. DNA was stained with Hoechst33342. Based on the Hoechst Blue fluorescence (i.e., DNA contents), the testicular cells can be classified as haploid cells (Round sperms and elongating sperms, RS/ES), diploid cells, and tetraploid cells (Leptotene or zygotene spermatocytes, L/Z; pachytene or diplotene spermatocytes, P/D). **d** A schematic diagram showing the effect of disruption of *Piwi1*, *Piwi2*, *Piwi3*, and *Piwi4* on male germ cell development in golden hamsters and mice.

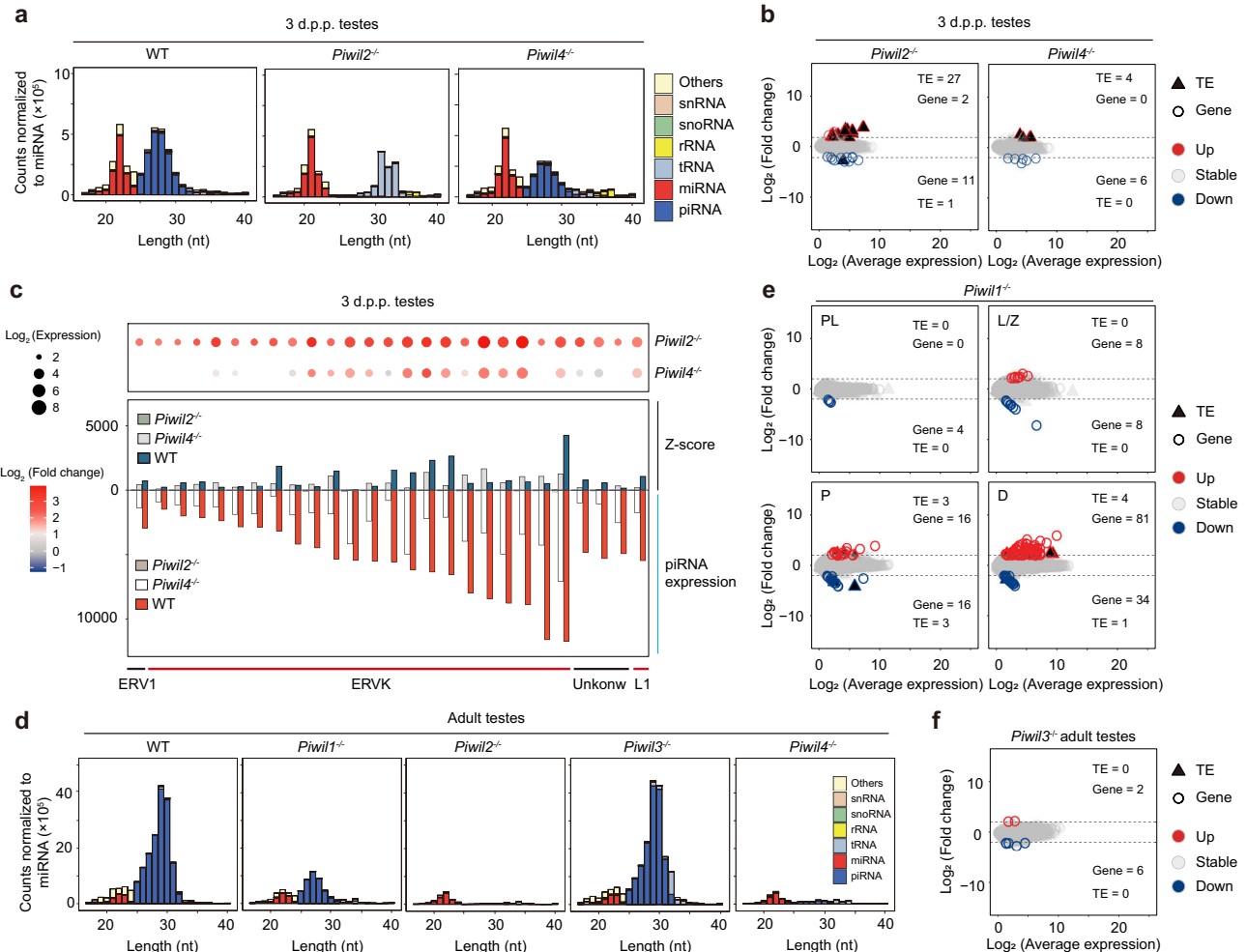

**Fig. 6 | Impaired TE silencing or gene expression in *Piwi*-deficient spermatogenic cells. a** The composition of WT or *Piwi*-deficient small RNA categories according to their length distribution in postnatal testes. Data are the average values of two biological replicates. **b** Analysis of differentially expressed consensus TEs and genes in WT versus *Piwi*-deficient testes. The expression levels of TEs or genes were normalized to exogenous ERCC RNA spike-in. The significantly up-regulated or down-regulated TEs and genes (≥four-fold; FDR < 0.01, permutation test) are indicated in red or blue, respectively. TE or gene are indicated by different shapes, and the TE or gene number is shown at the top. Data are the average values of three or four biological replicates. **c** Upper: Expression level changes of up-regulated TEs in different mutant testes, where circle size represents log₂-transformed expression, and circle color represents log₂-transformed fold-change compared to WT. Lower: The piRNA expression and Z-score statistics of up-

regulated TEs in WT, *Piwil2*⁻/⁻, or *Piwil4*⁻/⁻ postnatal testes. The upper panel exhibits Z-score, and the lower panel exhibits piRNA expression. The different genotypes are represented by different colors. Data are the average values of two biological replicates. **d** The composition of WT or *Piwi*-deficient small RNA categories according to their length distribution in adult testes. Data are the average values of two biological replicates. **e, f** Analysis of differentially expressed consensus TEs and genes in purified pre-leptotene (PL), leptotene/zygotene (L/Z), pachytene (P), and diplotene (D) spermatocytes of WT and *Piwil1*⁻/⁻ testes (**e**) or in *Piwil3*⁻/⁻ adult testes (**f**). TE/gene transcription levels were normalized to exogenous ERCC RNA spike-in. Significantly up- or downregulated TEs/genes (≥four-fold; FDR < 0.01, permutation test) are indicated in red or blue, respectively. TEs and genes are indicated by different shapes with TE or gene number shown at the top. Data are means of three or four biological replicates.

Supplementary Fig. 11a). Immunostaining showed that PIWIL4 was restricted to the cytoplasm and apparently unable to enter the nucleus in *Piwil2*⁻/⁻ postnatal testes, possibly attributable to the loss of bound piRNAs (Supplementary Fig. 11b). This observation, combined with the preferential 10 A site in PIWIL4 28-nt piRNAs found in WT testes, together supported the likelihood that PIWIL4 28-nt piRNA production was dependent on PIWIL2 and probably through heterotypic Ping-Pong between PIWIL2 and PIWIL4 proteins.

In contrast, in *Piwil4*⁻/⁻ 3 d.p.p. testes, PIWIL2-bound piRNAs were still present (Fig. 6a and Supplementary Fig. 11a) and PIWIL2 localization remained undisturbed (Supplementary Fig. 11c). However, the ratio of antisense piRNAs to total piRNAs and Ping-Pong signal was generally reduced in *Piwil4*⁻/⁻ gametes (Fig. 6c and Supplementary Fig. 11d). In addition, TEs were significantly upregulated in both *Piwil2*⁻/⁻ or *Piwil4*⁻/⁻ testes, but transcriptomic profiles did not significantly differ from WT (Fig. 6b, Supplementary Data 4 and 11). Interestingly, the vast majority

of the un-silenced TEs were identical between the two mutants, mostly belonging to the ERVK subfamily (Fig. 6c). In addition, greater up-regulation of TEs in *Piwil2*⁻/⁻ testes compared to *Piwil4*⁻/⁻ could potentially explain greater spermatogenic impairment in *Piwil2*⁻/⁻ gametes. We also detected an increasing Ping-Pong signal of some ERVK-derived piRNAs in *Piwil4*⁻/⁻ gametes (Fig. 6c), suggesting that PIWIL2 can perform homotypic Ping-Pong to partially compensate for the loss of heterotypic Ping-Pong between PIWIL2 and PIWIL4 in silencing some specific TEs. Taken together, these results highlighted the interconnected, but non-redundant roles of PIWIL2 and PIWIL4 in mutual regulation of piRNA production and subcellular PIWI localization during the first wave of spermatogenesis.

In adult testes, the disruption of *Piwil1* leads to a significant reduction in piRNA production, with only PIWIL2 27-nt piRNA retained in these mutants (Fig. 6d and Supplementary Fig. 11e). Pre-leptotene, leptotene/zygotene, pachytene, and diplotene spermatocytes from

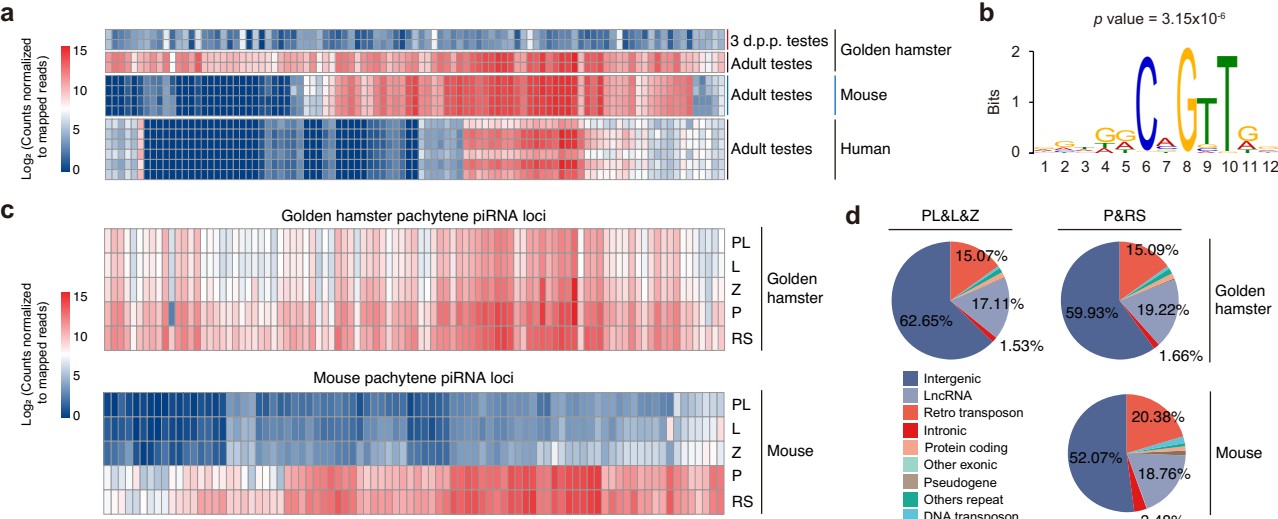

**Fig. 7 | Comparison of testicular pachytene piRNAs between golden hamsters and mice. a** Heatmap of piRNA abundance for the golden hamster pachytene piRNA loci and the syntenic loci in mouse and human. **b** MEME identification of an A-MYB binding motif in the promoter regions of pachytene piRNA loci of golden hamsters. The optimal enrichment *p* value of the motif is determined according to Fisher's exact test, adjusted for multiple tests using a Bonferroni correction. **c** Heatmap of piRNA abundance in the testes of golden hamster and mouse testes across developmental stages. In golden hamster, the 97 pachytene piRNA loci identified above are used with a corresponding set of well-defined pachytene piRNA loci from mice[48]. PL pre-leptotene, L leptotene, Z zygotene, P pachytene, RS round sperm. **d** Pie charts illustrating the genomic annotation of loci responsible for pachytene piRNAs in mouse and golden hamster male germ cells, including intergenic, intronic, other exonic, lncRNA, pseudogene, protein-coding, other repeats, DNA transposon, and retro transposon regions.

WT and *Piwil1* mutant testes were purified and RNA-seq were conducted to compare transcriptomes at the same stage. The results indicate that 4, 16, 32, and 115 genes are dysregulated in the pre-leptotene, leptotene/zygotene, pachytene, and diplotene *Piwil1* mutant spermatocytes, respectively (Fig. 6e, Supplementary Data 12). However, the number of dysregulated TEs was negligible (Supplementary Data 13), suggesting that PIWIL1 primarily participates in gene regulation rather than TE silencing. These findings align well with PIWIL1 bound piRNAs originating in intergenic regions (Supplementary Fig. 10e). Gene ontology (GO) analysis revealed that these differentially expressed genes (DEGs) were enriched in functions related to "sperm motility" and "spermatid development" (Supplementary Fig. 12), suggesting important roles in spermatocyte development and maturation. These findings highlight the crucial transcriptional regulatory role of PIWIL1 in spermatogenesis, even in leptotene and zygotene spermatocytes. In both *Piwil2*[−/−] or *Piwil4*[−/−] testes, almost all piRNA populations were lost, potentially due to the depletion of gametes in adult males. As a result, the detailed impacts of *Piwil2* and *Piwil4* deficiency on piRNA biogenesis in adult testes could not be reliably determined through sequence analysis. *Piwil3* deficiency had no discernible effect on smRNA, mRNA, or TE profiles in testes (Fig. 6d, f), in agreement with its normal spermatogenic phenotype.

**Pachytene piRNAs are generated before pachytene stage in golden hamsters**

To identify pachytene piRNAs in golden hamsters, we compared the abundance of piRNAs originating from the same genomic loci between 3 d.p.p. and adult testes. This analysis identified 97 loci responsible for producing piRNAs with ≥4-fold higher abundance in adult testes compared to 3 d.p.p. testes, which were thus designated as pachytene piRNA loci (Fig. 7a and Supplementary Data 14)[38]. Analysis of genomic synteny at these loci indicated that ~58.8% (57/97) of pachytene piRNA-producing genes in the golden hamster genome were found at locations syntenic with the mouse genome, and ~24.7% (24/97) were syntenic with the corresponding piRNA loci in the human genome (Fig. 7a and Supplementary Data 15). By contrast, ~38.1% (37/97) of the piRNA loci were rodent-specific, while another ~33.0% (32/97) were derived

from golden hamster-specific loci. These findings suggested that a substantial proportion of pachytene piRNA genes are syntenically conserved between golden hamsters and other mammals. Intriguingly, we also identified a conserved A-MYB-binding motif in the promoter region of several golden hamster pachytene piRNA genes (Fig. 7b). These findings supported that the production of precursor RNAs for pachytene piRNA is likely controlled by a specific set of transcription factors in golden hamsters, which is consistent with reports in mice, macaques, and humans[39,40].

Next, we analyzed the expression patterns of pachytene piRNAs derived from the loci identified above in smRNA-seq data from several developmental stages, which revealed that pachytene piRNAs were generated throughout the meiotic stage in golden hamsters (Fig. 7c and Supplementary Fig. 13). These findings sharply contrasted with the well-defined set of pachytene piRNAs produced in spermatocytes after entering the pachytene stage of meiosis in mice (Fig. 7c, d)[40]. These observations suggested that the regulatory functions of pachytene piRNAs may initiate earlier in golden hamsters than in mice, which is consistent with the phenotypic differences observed between *Piwil1*-deficient mice and *Piwil1*-deficient golden hamsters.

## Discussion

In this study, using golden hamsters as a model, we profiled the expression and sub-cellular localization of all four PIWI proteins, characterized their bound piRNAs, and investigated the developmental defects induced by their respective knockout in both males and females. This work provides insight into the physiological and pathological functions of PIWIs in mammalian reproduction and fertility.

The expression, localization, and function of PIWI proteins are tightly coordinated in golden hamsters to orchestrate the proper development of gametes and early embryos. In females, PIWIL2 is most likely responsible for maintaining oocyte quiescence, while PIWIL1 and PIWIL3 are involved in ensuring the developmental competence of oocytes. However, PIWIL1 primarily co-localizes with DDX4 in germinal granules or cytoplasmic foci and plays essential roles in silencing TEs and depleting maternal mRNAs, whereas PIWIL3 preferentially co-localizes with TDRKH on the mitochondrial outer membrane. Further,

PIWIL3 knockout in oocytes significantly reduces global DNA methylation but has little effect on either TE expression or transcriptomic profile[23,24]. Thus, these PIWIs appear to have distinct localization patterns and functions in oocytes, although PIWIL1 piRNAs can partially compensate for the loss of PIWIL3 piRNAs, suggesting that PIWIL1 can complement PIWIL3 function in regulating TEs and genes to some extent. Interestingly, in two-cell embryos, PIWIL1 enters nuclei, and PIWIL3 forms PIWIL1-like cytoplasmic foci, suggesting that PIWIL1 may alternate between cytoplasmic and nuclear roles while PIWIL3 transiently performs redundant cytoplasmic functions upon PIWIL1 translocation to the nucleus.

By contrast, in male postnatal testes, PIWIL4 piRNA production is dependent on heterotypic Ping-Pong signaling between PIWIL2 and PIWIL4, and thus PIWIL2 disruption not only results in the loss of PIWIL2 piRNAs but also impairs PIWIL4 piRNA generation and consequently, its nuclear localization, an effect also observed in mice[41]. However, PIWIL4 disruption does not negatively affect either PIWIL2 piRNA production or PIWIL2 localization, and numerous piRNAs are produced via increased homotypic Ping-Pong by PIWIL2, partially compensating for the loss of PIWIL4. In adult testes, PIWIL1 and PIWIL2 appear to function independently, and both are required for spermatogenesis. These findings highlight the distinct but tightly interconnected functions of PIWI proteins.

As possibly the most commonly used animal model, mammalian piRNA studies have most extensively studied mice. However, we and others have demonstrated that mice may not be an accurately representative model for piRNA studies in females[9,23–25]. While in males, our results indicate that the composition and function of PIWIs and piRNAs do not differ significantly between mice and golden hamsters. First, the expression profiles and subcellular localization of PIWI proteins in golden hamsters are consistent with those previously reported in mice[14–16,41–43]. Second, disruption of *Piwil1*, *Piwil2*, or *Piwil4* genes all result in complete male sterility and severe spermatogenesis defects in both golden hamsters and mice[14–16]. Third, interconnected roles of PIWIL2 and PIWIL4 have been defined in both mice and golden hamsters[41]. Finally, pachytene piRNA genes are primarily located in syntenically conserved loci and many of which are regulated by a common transcription factor[38–40].

However, there are also some noteworthy differences in the function of PIWIs between male mice and golden hamsters. In particular, individual knockout of *Piwil1*, *Piwil2*, or *Piwil4* leads to consistently more severe consequences in golden hamsters than those observed in the C57BL/6 mouse model commonly used in piRNA studies. Additionally, we also observed that pachytene piRNA expression begins earlier in golden hamsters than in mice, which may explain the greater severity of defects in spermatogenesis in *Piwil1*⁻/⁻ golden hamsters. Recently, several studies revealed that mutations in the piRNA pathway are also related to male fertility disorders in humans[44,45]. These similarities and species-specific differences in PIWI functions indicate that although either mice or golden hamsters can be selected as a model for investigating piRNA functions and mechanisms in male mammals, the results should be interpreted with strict caution, particularly regarding their generalizability to human fertility. In addition, the characterization of the relationship between PIWI and piRNA expression patterns in human germ cells in future work will facilitate the selection of appropriate models for the study of human fertility.

These cumulative findings expand the scope of our understanding regarding the biological function of the piRNA pathway in both male and female germ cells and provide in-roads for the development of potential treatments for human infertility.

## Methods

### Animals and sample collection
LVG Golden Syrian hamsters (*Mesocricetus auratus*) were purchased from Beijing Vital River (Charles River) and Liaoning Changshen Biotechnology. Animals were managed in a light cycle of 14L:10D (light:dark) or 12L:12D. All animal experiments were approved by Nanjing Medical University Institutional Animal Care and Research Committee (IACUC code:1709024). Growing, full grown and MII oocytes and early embryos before implantation (9 h.p.e.a., 33 h.p.e.a., 44 h.p.e.a., 52-54 h.p.e.a.) were collected as described previously[24].

### Generation of polyclonal antibodies for PIWIL3 and PIWIL4
Polyclonal antibodies used for immunoprecipitation were produced in rabbits by GL Biochem (Shanghai, China). A peptide with the sequence C-HGPASARRGGSQPPRAR was used as the antigen to produce the anti-PIWIL3 antibody. A peptide with the sequence C-RSPGDLTATSAT was used as the antigen to produce the anti-PIWIL4 antibody. Cysteine residues were included at the N terminal of the synthetic peptides for later coupling. The rabbits used for antibody production underwent seven rounds of immunization with the appropriate antigen prior to blood collection. Antibodies were then purified with peptide affinity chromatography.

### Immunoprecipitation assay on HEK293T cells
The specificity of anti-PIWI antibodies were tested by immunoprecipitation on HEK293T as described previously[7]. Briefly, HEK293T cells were transiently transfected with plasmids encoding Flag-tagged PIWIL1, PIWIL2, PIWIL3 and PIWIL4, respectively, by Lipofectamine 2000 (Invitrogen, 11668019) according to standard procedures. Cells were harvested 36 h after transfection and homogenized in lysis buffer (50 mM Tris-HCl [pH 7.4], 1% NP-40, 150 mM NaCl, 0.5 mM DTT, and 1× proteinase inhibitor cocktail (Sigma)) for 10 min at 4 °C. Then, the lysates were centrifuged at $12,000 \times g$ for 10 min at 4 °C. 400 μl of supernatant (40 μl were kept as input) was mixed with protein G beads specifically conjugated with 5 μg of PIWIL1, PIWIL2, PIWIL3 and PIWIL4 antibodies, respectively, and incubated overnight at 4 °C with gentle rotation. After washing the beads four times with a wash buffer consisting of 50 mM Tris-HCl (pH 7.4), 0.005% NP-40 and 150 mM NaCl, the IP pellets were boiled at 95 °C for 5 min with 1× protein loading buffer. Anti-Flag-HRP (Sigma, A8592, 1:5000) was used to detect the target proteins via western blot. Uncropped immunoblots are provided in the source data file.

### Generation of *Piwi*-deficient animals
sgRNAs were designed using CRISPRdirect (http://crispr.dbcls.jp/) and CRISPOR (http://crispor.tefor.net/). The fragments containing the T7 promoter and sgRNA were PCR-amplified and used as templates to produce and amplify sgRNAs in vitro transcription using the HiScribe T7 High Yield RNA Synthesis Kit (NEB, E2040S). The complementary target oligonucleotides were cloned into the pUC57-T7-sgRNA-trcRNA vector through the BsaI restriction site. *Cas9* gene tagged with a T7 promoter was PCR amplified from a pX330-U6-Chimeric_BB-CBh-hSpCas9 plasmid (Addgene plasmid 42230) and used as a template for in vitro transcription using the mMESSAGE T7 ULTRA Transcription Kit (Invitrogen, AMB1345). All sgRNAs and Cas9 mRNAs were purified by lithium chloride precipitation and stored at −80 °C.

Preparation of donor/recipient hamsters and microinjection were performed as described previously[24]. Briefly, two-cell embryos were obtained from the donor superovulated females mated with males and injected with 50 ng/μl Cas9 mRNA and 20 ng/μl sgRNA each under a microscope with a red filter in a red-light room. The injected two-cell embryos were cultured in mHECM9 + HSA medium under a humidified 10% $CO_2$ environment at 37.5 °C for 15−30 min, and transferred to the pregnant albino recipients, with 15−30 embryos for each recipient. Founder pups were mated with WT males and females to produce the F1 generation. Genotyping of F1 pups was performed using a one-step mouse genotyping kit according to the manufacturer's instructions (Vazyme, PD101−01). A list of the oligo sequences used is provided in Supplementary Data 1. All four Piwi mutant hamsters were backcrossed

for a minimum of nine generations before crossing heterozygotes to obtain homozygotes.

## Fertility analysis

WT or mutant female golden hamsters at the pro-estrus stage (2–4 months old) were caged overnight with WT or mutant males at a 1:1 ratio, and vaginal sperm was examined using a microscope the following morning. The pregnancy rate was calculated according to the number of sperm with successful deliveries per observation in the vagina and no sticky vaginal discharge appeared during the next two estrous cycles.

## Flow cytometric analysis and FACS

The testes from 6-week-old or 16-week-old hamsters were dissociated to prepare a single-cell suspension. Spermatocytes were stained with Hoechst 33342 for FACS based on the previously described method[37] with minor modification. Briefly, after removing tunica albuginea, each testis was digested with 400 U/mL Collagenase IV and 100 μg/mL DNase I for 4 min at 35 °C, then the tubes containing tubules were allowed to stand vertically for 1 min to remove the interstitial testicular cells. The seminiferous tubule was then digested using 0.25% trypsin, 100 U/mL Collagenase IV, and 100 μg/mL DNase I for 8 min at 35 °C. After the digestion, fetal bovine serum (10% final) was added to inactivate the trypsin, and the resulting suspension was passed through a 70 μm cell strainer and stained with Hoechst 33342 (6 μg/million cells) for 30 min. Finally, the suspension was passed through a 40 μm nylon cell strainer. The cell suspension was kept on ice and was stained with 10 μL PI (1 mg/mL) at room temperature prior to sorting.

Data analysis was done using a BD FACSAria and Flowjo 10.4 software. Hoechst was excited using a 355 nm laser, and the wide emission spectrum of the dye was detected in two distinct channels: the "Hoechst Blue" (450/40 nm band-pass filter) and the "Hoechst Red" (670 nm long-pass filter). The latter was also used to detect PI. A dichroic mirror (610 nm long pass filter) was used to split these emission wavelengths. Forward Scatter (FSC-A) and Side Scatter (SSC-A) were detected using a 488 nm laser. Four-way sorting was performed using a 70-micron nozzle size. The sorting flow rate was adjusted to 2000–4000 events per second. A minimum of 100,000 events, were pre-recorded before the setting of gates. 40,000 cells were sorted into 5 mL polypropylene round-bottom tubes containing 1 mL of 10% FBS in DMEM.

## Meiotic spreads and immunostaining

Nuclear spreads were performed as described previously[37] with minor modifications. Briefly, a 40 μL aliquot of sorted cells was mixed with 80 μL hypotonic buffer (30 mM Tris-Cl, 50 mM sucrose, 17 mM sodium citrate dehydrate, 5 mM EDTA, 0.5 mM DTT, 1 × CK) and incubated at RT for 40 min. Cells were pelleted at $400 \times g$ for 4 min, the supernatant was removed, and cells were re-suspended with 80 μL of 0.1 M sucrose (pH, 8.2). Meanwhile, using a hydrophobic barrier PAP pen (VECTOR, H-4000), four $10 \times 10 \text{ mm}^2$ squares were drawn on a clean glass ($25 \times 75 \text{ mm}^2$, CITOTEST, China). Each square was dipped in 1% paraformaldehyde (pH, 9.2) supplemented with 0.1% Triton X-100. 40 μL hypotonic cells suspended with 0.1 M sucrose were then pipetted onto the squares and the slides were placed in a slightly opened humid box overnight. Next, the slides were washed once with 0.2% Photoflo (Kodak) for 1 min and followed by serial washes with DPBS containing 0.5% Triton X-100, DPBS containing 0.05 %Triton X-100, and DPBS. Slides were treated with blocking buffer (3% BSA, 5% donkey serum, and 0.05% Triton X-100) for 60 min at room temperature and then incubated overnight at 4 °C with primary antibodies (anti-SCP3, 1:750, Abcam, #ab15093; anti-SCP3, 1:500, SANTA CRUZ, #74569; anti-SCP1, 1:500, Abcam, #ab15090; anti-phospho-Histone cH2A.X, 1:1000, Millipore, #05-636) in blocking buffer. After washes, slides were incubated with secondary antibodies (donkey anti-rabbit cy3, 1:400, Jackson ImmunoResearch, #711-165-152; donkey anti-mouse 488, 1:800,

Molecular Probes, #A-21202) in blocking buffer supplemented with 1 μg/mL 4,6-diamidino-2-phenylindole (DAPI) for 60 min. Slides were subsequently washed as described above, mounted in Vectashield (Vector Laboratories), and sealed with clear nail polish. Fluorescence was observed using an Olympus SpinSR spinning disk confocal super-resolution microscope.

## Immunoblot analysis

Testes of 6–8 weeks males and MII oocytes of 8–10 weeks females were lysed in ice-cold RIPA buffer or M-PER reagent (Thermo Scientific, #78503) containing protease inhibitors. A capillary-based immunoassay was performed using the Wes-Simple Western method with the anti-rabbit and anti-mouse detection module (ProteinSimple, #SM-W004, #DM-001, #DM-002). Protein expression was measured by chemiluminescence and quantified as an area under the curve using the Compass for Simple Western program (ProteinSimple). Proteins were detected with the following primary antibodies: PIWIL1 (1:100; GL Biochem, China); PIWIL2 (1:100; Millipore, #MABE363); TUBA (1:200; Proteintech, #66031-1-Ig); PIWIL3 (1:100, lab produced) and PIWIL4 (1:100, lab produced); Beta Actin (1:200, Proteintech, #66009-1-Ig).

## Histological analysis

Testes and ovaries were fixed in Bouin's fixative (Sigma-Aldrich, MBD1105) overnight at 4 °C, embedded in paraffin, and sectioned at 5μm. For periodic acid–Schiff (PAS, Solarbio, #G1281) staining, sections were deparaffinized and rehydrated and then stained with PAS. Slides were mounted in neutral resins and images were acquired using an Axio Scan.Z1 Digital Slide Scanner (Zeiss).

## Immunofluorescence analysis

Testes and ovaries were fixed in 4% paraformaldehyde (PFA) overnight at 4 °C followed by embedding in paraffin and sectioning at 5 μm. Sections were deparaffinized and rehydrated with citrate buffer (pH, 6.0) for antigen retrieval. After blocking with 2% bovine serum albumin and 4% normal donkey serum in PBS overnight at 4 °C, the sections were treated with primary antibodies for 2 h at 25 °C. The slides were washed twice with PBST and then treated with the secondary antibody or PNA–FITC (Sigma-Aldrich, L7381, 1:1000) for 1 h at 25 °C. For TUNEL staining, 4% PFA fixed testes sections were stained using the In Situ Cell Death Detection Kit (Roche, 12156792910) according to the manufacturer's instructions. After washing twice with PBST, the sections were stained with 1 μg/mL DAPI (Sigma-Aldrich, HT10132). Fluorescent images were captured using the Olympus SpinSR spinning disk confocal super-resolution microscope.

Oocytes and embryos were collected in vitro or in vivo and fixed with 4% PFA, permeabilized with 0.2% Triton X-100 in PBS, and then blocked with PBS containing 1% bovine serum albumin and 2% normal donkey serum overnight at 4 °C. The oocytes and embryos were incubated with primary antibodies for 2 h at 25 °C. After washing twice with PBST, samples were further incubated with secondary antibodies for 1 h at 25 °C. Nuclei were stained with DAPI. Fluorescent images were captured using the Olympus SpinSR spinning disk confocal super-resolution microscope.

Proteins were detected with the following primary antibodies: anti-PIWIL1 polyclonal antibodies (1:200, lab produced)[7]; anti-PIWIL2 monoclonal antibodies (1:200, Millipore, MABE363); anti-PIWIL3 polyclonal antibodies (1:200, lab produced); anti-PIWIL4 polyclonal antibodies (1:200, lab produced); anti-DDX4 (1:200, Abcam, ab27591); anti-TDRKH (1:80, Thermo Scientific, PA5-48098); anti-DCP1A (1:200, Santa Cruz, sc-100706), anti-DDX6 (1:100, Santa Cruz, sc-376433), anti-ATP5A1 (1:200, Proteintech, 66037-lg), anti-TDRD1 (1:50, R&D, MAB6296), anti-LIN28 (1:200, CST, 8641), anti-SCP3 (1:500, Abcam, ab15093) or anti-γH2AX (1:1000, Millipore, 05-636); anti-tubulin FITC antibodies (1:500, Sigma-Aldrich, F2168). Secondary antibody information is as follows: Alexa488-conjugated donkey anti-mouse IgG

(1:800, Molecular Probes, A-21202); Cy3-conjugated donkey anti-rabbit IgG (1:400, Jackson ImmunoResearch, 711-165-152); Cy3-conjugated donkey anti-rabbit IgG (1:400, Jackson ImmunoResearch, 713-165-147); Alexa488-conjugated Goat anti-rabbit IgG (1:500, Thermo Fisher, A-11034); Alexa594-conjugated donkey anti-rabbit IgG (1:200, Thermo Fisher, A21207); CF647-conjugated donkey anti-rabbit IgG (1:1000, Sigma, A37573).

## Confocal microscopy, image acquisition, and image processing

Images were acquired on an Olympus SpinSR spinning disk confocal super-resolution microscope using lens as bellows: 20X (Numerical Aperture = 0.8, air immersion, no correction collar, total system magnification = 20X, sampling rate of signal: XY = 325 nm/pixel); 40X (Numerical Aperture = 0.95, air immersion, correction collar used, total system magnification = 40X, sampling rate of signal: XY = 162.5 nm/pixel); 60X (Numerical Aperture = 1.42, oil immersion, correction collar used, total system magnification = 60X, sampling rate of signal: XY = 108.3 nm/pixel); 100X (Numerical Aperture = 1.5, oil immersion, correction collar used, total system magnification = 100X or 320X, sampling rate of signal: XY = 65 nm/pixel or XY = 20.3 nm/pixel). Four-color laser excitation, including 405 nm ≥ 100 mW; 488 nm ≥ 100 mW; 561 nm ≥ 100 mW; 640 nm ≥ 100 mW, and the corresponding emission band-filter sets are 447/60 (DAPI), 525/50 (GFP), 617/73 (RFP), 685/40 (CY5), respectively. XY resolution of the camera (Hamamatsu, ORCA-Fusion) is 2304 × 2304 pixels. The laser power and exposure time were set in an antibody-dependent manner to 5 to 20% and 100 to 400 ms, respectively, below saturation condition. Acquired images were not manipulated except for linearly adjusting of contrast and brightness. All figures were made using FIJI or Adobe Photoshop CC 2022.

## Transmission electron microscopy analysis

Ovaries were fixed in 2% PFA + 2.5% glutaraldehyde for 40 min at room temperature, washed twice with PBS, then treated with 0.2% tannic acid for 1 min. After fixation, samples were dehydrated in sequential gradients of ethanol (30%–50%–70%–80%–95%–100%). Ethanol was replaced with acetone, followed by resin infiltration, embedding, and polymerization. Ultra-thin sections of 50 nm were performed on the embedded tissue, and suitable sections were selected by pre-staining with toluidine blue. Ultra-thin sections were transferred to copper mesh, stained with uranyl acetate and lead citrate, washed with distilled water, dried, and photographed using a transmission electron microscope (Zeiss).

For detecting PIWIL3 expression, ovaries were fixed at 4 °C overnight in PBS containing 4% paraformaldehyde, 0.2% picric acid, and 2% sucrose, washed three times with PBS, dehydrated with ethanol, embedded into LR Gold, and polymerized with UV at −25 °C for 48 h. Ultra-thin sections (50–70 nm) were cut, collected on parlodion coated grids, blocked with 3% BSA, stained with anti-PIWIL3 (1:200, lab produced) and gold particle-conjugated secondary antibodies (1:50, Aurion), counterstained with uranyl acetate, and visualized with an FEI Tecnai G2 Spirit electron microscope.

## RNA immunoprecipitation

RNA immunoprecipitation of PIWI proteins in oocytes and testes was performed as previously described[24]. Protein A Magnetic Beads (NEB, S1425S) were washed twice with RIP lysis buffer composed of 50 mM Tris-HCl (pH, 7.4), 150 mM NaCl, 1 mM EDTA, 0.5% NP-40, 0.5 mM DTT, 0.1 U/μL RNase Inhibitor (Thermo Scientific, EO0381) and 0.4 U/μL Proteinase inhibitor cocktail (Sigma). Every 1.5 μg of indicated antibody and 2.5 μL original beads slurry were incubated in 10 μL RIP lysis buffer on a rotator at room temperature for 0.5 h. Coupled beads were washed with RIP buffer before use. 10 MII oocytes were pooled and lysed by adding 30 μL cold RIP lysis buffer in a DNA low-bind tube (Eppendorf, Germany) on ice for 10 min, respectively, followed by centrifugation at 12,000 g for 5 min at 4 °C. Each 7.5 μL oocyte lysis was

transferred into a tube that contained the antibody-coupled beads for RIP assay. The oocyte lysis left in the tube after distribution for IP reactions was treated with TRIzol to extract RNAs as input. For IP reactions, after rotating for 5 h at 4 °C, the beads were washed with RIP lysis buffer with rotation at room temperature for 5 min, and washing was repeated three times, followed by another wash with PBS. The beads were resuspended by 2 μL water and incubated at 75 °C for 5 min then cooled on ice, the supernatants were recovered on the magnetic holder and prepared for library construction.

## Small-RNA library construction and sequencing

Small RNA libraries were constructed as previously described[24]. Briefly, extracted RNA or cell lysates were incubated at 72 °C for 3 min and subjected to 3′ adapter ligation. Lambda exonuclease and 5′ dead-enylase were used in combination to remove the excess 3′ adapter. 5′ adapter ligation and reverse transcription reaction followed. Pre-amplification was performed, and 1 μL of the product was used for final amplification. The amplified libraries were separated on a 6% poly-acrylamide gel, and 130–160 bp DNA fragments were selected. All libraries were sent for sequencing on an Illumina HiSeq X Ten platform or NovaSeq 6000.

## mRNA library construction and sequencing

mRNA libraries were constructed based on Smart-seq2 as previously described[24]. Briefly, extracted RNA or cell lysates were incubated at 72 °C for 3 min, followed by reverse transcription using ProtoScript II Reverse Transcriptase (NEB) and pre-amplification. The amplified cDNA was then purified by Agencourt AMPure XP beads (Beckman Coulter) and 2 ng of cDNA was used for the next tagmentation reaction. After final amplification, mRNA libraries were purified by Agencourt AMPure XP beads and analyzed on a 2200 Tapestation System (Aligent Technologies, Santa Clara, CA) for quality control. All libraries were sequenced 2 × 150 bp on an Illumina HiSeq X Ten or a NovaSeq 6000 platform (Illumina, USA).

## Categorization of small RNAs

Raw read1 fastq file was used to identify small RNAs, which were pre-processed using the FASTX Toolkit 0.0.14. The preprocessing steps are as follows: intercept the first 50 bp of reads and filter out low-quality sequences (-q 20, -p 90); After quality-filtering, sequencing reads were clipped from the 3′adapter allowing for a minimum match of 10 nt from the 5′ ends. Reads unable to match the adapter sequence or with lengths shorter than 17 bp were discarded, and the remaining reads were mapped to genome by bowtie ((hamster: BCM_Maur_2.0, mouse: mm10), bowtie version 1.2.1.1, parameters -k = 100 -v = 0). If the reads mapped more than 100 loci to the genome, 100 of all mapped loci are randomly output. The reads were aligned sequentially to known miRNA, tRNA, rRNA, snoRNA, and snRNA sequences by Bowtie without any mismatch allowed (bowtie parameter is -k = 100 -v = 0 -norc) and reads that aligned to pre-miRNA were identified as miRNAs. The remaining reads of the specified length were extracted and used to identify piRNAs according to the previously described method[46] with slight modifications. The length of candidate piRNAs in male samples is 25-32nt, and the length of candidate piRNAs in female samples is 17-32nt. The clustering parameters were determined as MinReads = 5 and Eps = 2500 bp by running a series of k-dist analyses of our data with different Eps and MinReads. All candidate clusters that satisfied these parameters were considered piRNA clusters, and the remaining sequences located in these clusters were defined as piRNAs without any further filtering. Reads in each library were normalized by sequencing depth or read counts of miRNAs or read counts of exogenous spike-in.

## Small-RNA spike-in information

Three unmethylated and three methylated small-RNA spike-ins were synthesized (Integrated DNA Technologies, IDT) and pooled. Spike-ins

of 8.14 × 10⁻⁵ pmol were added to the reaction for single oocytes or embryos or testes. The sequence of each spike-in can be found in our previous study[7].

## piRNA genomic annotation

For different subgroups of TEs or genes, piRNA reads in the sense and antisense direction were counted in order as retrotransposon (included ERV, other LTRs, LINE/L1, retrotransposon/L1-dep, LINE/L2, SINE/Alu, SINE/B2, SINE/others), DNA transposon, other repeats, protein-coding exon, pseudogene, lncRNA, other exonic, intron, or intergenic. If the piRNA overlapped with the annotated region at the genome level, it was considered to be derived from this region.

## Expression profiling for consensus TE-derived piRNAs

The piRNA reads were mapped to consensus TEs by bowtie, allowing up to 2-nt mismatches (using parameters: -a -v 2), and the expression levels of each TE-derived piRNA were normalized by exogenous spike-in or sequencing depth. If a piRNA could be mapped to different members of a TE subfamily, this piRNA was counted for each TE member. The same piRNA of each TE member was counted only once.

## Identification of ping-pong signaling

Ping-pong Z-score was calculated as described earlier[47]. Briefly, the Z-score is the difference of the number of piRNAs complementary to each other at a 5′-to-5′ distance of 10nt and the mean number of piRNAs at background 5′-to-5′ distances (defined as distances of 0–9 and 11–20 nt), divided by the standard deviation of the numbers of piRNAs at background distances. Alignments of piRNA to consensus TEs were performed with two mismatches allowed and used to interrogate TE-specific ping-pong Z-scores. If the piRNAs Z-score on TE is ≥1.96 (corresponding to $p$ value ≤ 0.05), it was concluded that the TE has a significant ping-pong signal.

## Data analysis of single-cell RNA-sequencing and differential expression analysis

The raw paired-end fastq reads had adapters removed and were quality-filtered by TrimGalore-0.5.0 with the parameters --paired --quality 20 --phred33 --stringency 1 --length 35. The remaining reads were mapped to the genome or identified TE by STAR v2.9. Uniquely mapped reads and multiple mapped reads with fewer than 10,000 genome copies were used for downstream analysis. The specific alignment parameters are as follows: --winAnchorMultimapNmax 10000 --outFilterMultimapNmax 10000 --alignSJoverhangMin 8 --alignSJDBoverhangMin 1 --outFilterMismatchNmax 999 --alignIntronMin 20 --alignIntronMax 1000000 --alignMatesGapMax 1000000 --twopassMode Basic --outSAMprimaryFlag AllBestScore. Calculation of the read count of known genes or TEs was performed by feature-Counts v1.6.4 with the parameters -p -B -C -M -Q 20. The expression of known genes or TEs was normalized to the ERCC or mapped reads. Genes or TEs with normalized count >2 in 40% or more of samples were used for differential analysis. Differentially expressed genes or TEs were identified using the permutation test and the derived $p$ values were adjusted for multiple testing using the Benjamini-Hochberg procedure (≥4-fold change and FDR < 0.01).

## De novo identification of TEs in the golden hamster

Prediction and classification of TEs by RepeatModelerV2.0 was performed with the parameters -LTRStruct. Annotations of all predicted representative TEs were established in the golden hamster genome (BCM_Maur_2.0) by RepeatMasker v4.1.1 using the parameters -e crossmatch -s -a and -lib.

## Identification of genomic transposon activity

The nucleotide substitution rates of all consensus TEs in golden hamsters were calculated according to the previously described method[25](https://github.com/fhorvat/bioinfo_repo/tree/master/papers/piRNA_2021). According to our test, we found that the base substitution rate of most expressed TEs was lower than 0.15, and a threshold criterion was set: if the substitution rate of a TE < 0.15, then this TE was potentially active.

## Gene ontology annotation analysis

Gene ontology annotation analysis by the clusterProfiler package in R. The mouse molecular function (MF) and biological process (BP) gene sets were used for hamsters and the simplifyEnrichment package was used to simplify the analysis results.

## Definition of pachytene piRNA clusters

We defined golden hamster pachytene piRNA clusters as those whose piRNA cluster abundance was ≥4-fold lower in the 3 days post-partum testis tissue of hamster than that of adult golden hamster. Mouse pachytene piRNA gene annotation were downloaded from previously reported[48].

## Synteny analysis

We conducted an evolutionary conservation analysis of pachytene piRNA clusters in the golden hamster using a similar approach as previously described for human piRNA clusters[48]. We mapped the 97 hamster pachytene piRNA clusters to the human and mouse genomes using UCSC chain files and liftOver, with the parameter -minMatch=0.1, and recorded the syntenic regions of pachytene piRNA clusters that could be lifted over to each of the other species. The coordinates of the syntenic region in the other species which overlapped a hamster pachytene piRNA clusters on the same genomic strand were adjusted to be the piRNA cluster coordinates we annotated in that species. To include piRNAs that mapped to the boundaries of the syntenic regions in the other species but did not overlap with piRNA clusters on the same genomic strand, we extended these syntenic regions by 10 kb on both ends. Finally, we calculated the piRNA abundance in these regions using small RNA-seq data in that other species.

## Motif discovery

We used transcriptome data to identify the transcription start sites (TSS) of pachytene piRNA clusters. The promoter region was defined as the region between −300 and 200 bp surrounding the TSS. Sequence motifs in these putative promoter regions were detected using the MEME-AME tool[49,50].

## Sources of annotations

Golden hamster genome annotation (BCM_Maur_2.0) was downloaded from NCBI (https://www.ncbi.nlm.nih.gov/assembly/GCF_017639785.1), Gene annotations were also downloaded from NCBI. Mouse genome and Gene annotations were downloaded from UCSC. For golden hamster, functional RNA was retrieved from the following databases: miRNAs, sequences from a previous study[24]; tRNAs, RNACentral v17 (https://rnacentral.org/); 5 S and 5.8 S rRNA, snoRNAs and snRNA from Ensembl (https://asia.ensembl.org/index.html). For mice, miRNAs from miRbase v18; 18S, 28S, and 45S rRNA were downloaded from NCBI (https://www.ncbi.nlm.nih.gov/nucleotide/); tRNA was retrieved from gtRNAdb (http://gtrnadb.ucsc.edu/); Others were retrieved the same way as in golden hamster. Moreover, to fully classify the functional RNA-derived small RNAs, we combined the known functional RNA sequences annotated in mouse and rat genomes with related annotations in the golden hamster.

## Statistics and reproducibility

No statistical method was used to predetermine the sample size. The sample size was chosen on the basis of previous experience and similar studies to reliably measure experimental parameters according to standards in the relevant field. No data were excluded from the

analysis. All experiments were performed in at least three biological replicates, unless otherwise specified in the figure legends. The investigators were blinded to group allocation during constructing sequencing libraries. No other blinding was involved, but randomization was used. Otherwise, all data analyses were performed using unbiased software programs (PRISM (v.9, GraphPad)). Data are presented as mean ± s.e.m. The statistical tests and $p$ values are indicated in the figures or figure legends.

### Reporting summary
Further information on research design is available in the Nature Portfolio Reporting Summary linked to this article.

## Data availability
All deep-sequencing data have been deposited at the National Center for Biotechnology Information NCBI Gene Expression Omnibus (GEO) (http://www.ncbi.nlm.nih.gov/geo/) database under accession number GSE217621. The following publicly available datasets were used: human adult testis data GSE135791; mouse adult testis data PRJNA421205; mouse spermatogenesis data GSE101933; $PiwiI1^{-/-}$ related data GSE169528. Source data are provided with this paper.

## Code availability
There was no new code generated in this study. All code or software used is publicly available and have been specified in the "Methods" section.

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

## Acknowledgements

We thank the help of the staff at the Animal Core Facility of CEMCS. We thank all members of the Dr. Ligang Wu laboratory and Dr. Jianmin Li laboratory; Dr. Minghan Tong and Dr. Moufang Liu for their discussions and support for this project; Y. Xu for his assistance with high-performance computing; staff at the HPC storage and network service platform of CEMCS for supplying the computing resources. We thank Jiahao Sha, Ke Zheng, You-Qiang Su and Nan Ye (Nanjing Medical University, China) for their comments and suggestions. This work was supported by the following funding: the National Key R&D Program of China (2022YFA1303301 and 2021YFA1100201) and the National Natural Science Foundation of China (31970607 and 31470781) to L.W.; the National Key R&D Program of China (2021YFF0702500) and the National Natural Science Foundation of Jiangsu Province (BE2019730) to J.L.; National Natural Science Foundation of China (32200696) and the China Postdoctoral Science Foundation (2021M700146) to H.Z.

## Author contributions

L.W. and J.L. conceived and designed the study. J.L., A.S., Y.L., and W.Z. conceived and developed the methodology for genome editing of golden hamsters. W.Z., Y.L., Z.Z., C.Q., and Qh.C. generated the *Piwil1*^-/-^, *Piwil2*^-/-^, *Piwil3*^-/-^ and *Piwil4*^-/-^ golden hamsters. X.L., L.H., D.W., Y.S., Z.Z., Q.C., S.L., C.Q., R.Q., Y.G., and A.S. analyzed the phenotype of *Piwil1*, *Piwil2*, *Piwil3* and *Piwil4* mutants. X.L., D.W., and X.S. analyzed the spatio-temporal expression of all four PIWI proteins in gametogenesis and early embryogenesis. H.Z. and X.L. construct small-RNA and RNA libraries. X.L. and L.H. construct PIWIL1-, PIWIL2-, PIWIL3- and PIWIL4-IP libraries. W.X. performed bioinformatics analyses. X.L., H.Z., W.X., J.L. and L.W. interpreted the data of the experiments and wrote the manuscript.

## Competing interests

The authors declare no competing interests.
