## [Peer Review File · Nature Communications]

The non-redundant functions of PIWI family proteins in gametogenesis in golden hamstersREVIEWER COMMENTS

Reviewer #1 (Remarks to the Author):

In this beautiful manuscript, the authors describe comprehensively the piRNA pathway in golden hamsters. Using knockouts for all four PIWI proteins, they describe their distribution in different developmental stages, sub-cellular localization, phenotypes and small RNA populations. They compare their findings to the decades of information available from study of mice, the other major mammalian model in the field. Recently a couple of other publications have reported on the piRNA pathway in hamsters, but this manuscript completes this series to encapsulate everything into one manuscript.

There are several key findings that are of interest to the field. The relevance of piRNA pathway for the female germline (which is not the case in mice), compensation between PIWI proteins during transposon silencing, role of PIWIL3 (a fourth protein not found in mice), relevance of piRNA pathway for embryonic development (not seen in mice). This manuscript will be a reference for the community to access and cite. I like the presentation of the figures- very clear and self-explanatory. In fact, one can read the paper just by looking at the figures. I commend the authors for this heroic effort in putting this manuscript together.

Minor suggestions.

1. The terms pr-piRNAs and pa-piRNAs are unnecessary. Just adds to the confusion.
2. The text constantly refers to mouse models while describing the results from hamster. I think it is a bit confusing and distracting. Just describe from start to finish data from hamsters. Only in the discussion, compare the two models.
3. There are a lot of repetitions in the text due to the mouse-hamster comparison.

Reviewer #2 (Remarks to the Author):

Review uploaded as PDF.

Manuscript No.: 405166

"The non-redundant functions of PIWI family proteins in gametogenesis in golden hamsters"

Xiaolong Lv, Wen Xiao, Yana Lai, Zhaozhen Zhang, Hongdao Zhang, Chen Qu, Li Hou, Qin Chen, Duanduan Wang, Yun Gao, Yuanyuan Song, Xinjia Shui, Qinghua Chen, Ruixin Qin, Shuang Liang, Wentao Zeng, Aimin Shi, Jianmin Li, Ligang Wu

Wu and colleagues have characterized the expression, localization, and loss-of-function phenotypes for the four PIWI proteins in male and female golden hamsters. The study design, the experiments, and results give convincing evidence of the importance of the four PIWIL proteins in gametogenesis in hamsters, but many of their conclusions go too far beyond the existing evidence. Additionally, more detail is required regarding sequencing and genetics to properly assess the study, and several key experiments are missing statistical analysis or appear to have failed to correct for multiple hypothesis testing.

Specific Concerns

- (1) How did the authors determine that PIWIL2 was present in IMC and PIWIL4 was localized to the piP-body? What markers were used to identify mitochondria and piP bodies? Please show the data that supports the colocalization of PIWIL2 and PIWIL4 with these subcellular structures in testis and of PIWIL3 with mitochondria in four-cell embryos.
- (2) For how many generations were the knockout hamsters backcrossed to eliminate off-target CRISPR edits before the heterozygotes were crossed to obtain homozygotes for phenotypic and fertility analyses?
- (3) How were small RNAs identified as piRNAs? For PIWIL1,2, or 4, which are methylated, a small RNA can be categorized as a piRNA if it is enriched in an oxidized small RNA sequencing library or co-immunoprecipitated with the PIWI protein; for PIWIL3, co-immunoprecipitation is the only way to identify piRNAs.
- (4) What supports the claim that additional 29 nt piRNAs are made in *Piwil3*^{-/-} ovaries to compensate for loss of PIWIL3? Isn't it more likely that PIWIL3 and PIWIL1 compete for a common set of pre-piRNAs and that in the absence of PIWIL3, these are loaded into PIWIL1 and trimmed to 29 nt instead of 19 nt? Is the amount of PIWIL1 protein greater in *Piwil3*^{-/-} than in wild-type?
- (5) The authors make two apparently contradictory claims: first, that in *Piwil1*^{-/-} germ cells die before the pachytene stage and, second, that *Piwil1*^{-/-} testes are enriched for spermatocytes arrested at the pachytene or diplotene stages. I don't understand how both can be true.
- (6) Given that the distribution of cell types in *Piwil1*^{-/-} testes differs from wild-type, how can the authors be sure that the 1790 genes whose mRNA abundance increased in the mutant testes is caused by a loss of piRNA-directed regulation rather than the mismatch in tissue composition? To establish a change in mRNA

abundance caused by loss of piRNA function, purified germ cells must be compared to each other (e.g., FACS-purified primary spermatocytes from *Piwi1*^{-/-} vs. wild-type).

(7) The specific genes and transcription factors responsible for producing pachytene piRNA precursor RNAs have been defined in mice, macaque, and humans. Do the pachytene piRNAs in hamsters come from loci syntenic to those genes? From rodent- or hamster-specific loci?

(8) The claim that in hamsters there is no clear difference between pre-pachytene and pachytene piRNAs is semantics, not biology. Operationally, male mouse germ cells make transposon-silencing piRNAs, 3'-UTR-derived piRNAs, and "pachytene" piRNAs derived from a defined set of genes whose lncRNA transcripts are depleted of active transposons. For example, in male mice, the L1 transposon is desilenced in *Miwi* (*Piwi1*) mutants after the onset of meiosis. But the piRNAs involved do not derive from any of the pachytene piRNA loci and are instead derived from loci containing sequences antisense to L1. Despite the awkward nomenclature, "pachytene" piRNAs are defined by their source loci not their production at the pachytene stage of meiosis. The most evolutionarily conserved (by synteny not sequence) pachytene piRNA genes are also transcribed by a common set of transcription factors, A-MYB (*MYBL1*) and *TCFL5*. Given that the pachytene piRNA genes responsible for the majority of human "pachytene" piRNAs are found at syntenic locations in humans and are also transcribed by A-MYB and *TCFL5*, it is highly unlikely that this class of piRNAs (regardless of its name) is different in hamsters. Given the recent findings by Choi et al. (*PLoS Genet* 2021) and Wu et al. (*Nat Genet* 2020) that the majority of pachytene piRNAs have no detectable function in mice, it is unlikely that the composition and functions of small RNAs obviously differ significantly among male placental mammals. Similarly, the current manuscript provides little evidence that "the smRNA composition and function in male germ cells are likely diverse in different mammals."

(9) The claim that the consequences of individual knockout of *Piwi1*, *Piwi2*, or *Piwi4* are consistently more severe in golden hamsters than in mice may simply reflect the use of one specific inbred strain (C57BL/6) for the mouse studies. This claim needs to be softened.

Important Data Concern

Extended Data Figure 1 appears to be a composite of multiple gels. This type of manipulation is unlikely to meet the journal standards. Please provide the uncropped gels used to assemble the figure.

Other Points

(1) Figure 2A provides a far better demonstration of antibody specificity than Extended Data Figure 1A. Could 2A be moved to Figure 1?

(2) In addition to reference 23, Loubalova et al. (*Nat Cell Biol* 2021) should be cited for the female subfertility of *Piwi3* mutants.

(3) Please provide the following information for the light microscopy methods:

- Objective (magnification, N.A., immersion used, correction collar yes/no);
- Filter sets and excitation sources and wavelengths; if known used power and excited field of view;
- Camera (manufacturer, model, pixel size and used settings);
- Total system magnification and sampling rate of signal.

(4) In *Piwi3* mutant oocytes, what data show a geometric increase in 29 nt piRNAs bound to *PIWIL1*? Why is the increase geometric?

(5) In Extended Data Figures 5A, 5E, and 6C were the data corrected for multiple hypothesis testing. If not, the p-values need to be adjusted to take this into account.

(6) "Crosstalk" means the "unwanted transfer of signals between communication channels." The standard term for ping-pong between different PIWI proteins is heterotypic ping-pong; ping-pong within a single type of PIWI protein is homotypic ping-pong.

(7) Pachytene piRNAs in mice have been extensively characterized in both staged whole testes (Li et al., *Mol Cell* 2013) and FACS purified cells (Gainetdinov et al., *Mol Cell* 2018; Wu et al., *Nat Genet* 2020; Yu et al., *RNA* 2023), not simply in adult total testes.

(8) The report that "spermiRs" (not a term generally used by the field) are a very high proportion of pre-meiotic small RNAs (ref. 39) is not consistent with other, more quantitative analyses of spermatogonial miRNAs and piRNAs. Unless the authors can further support these claims using publicly available sequencing data performed with small RNA spike-ins and purified spermatogonia, the claim that there is a difference between mice and hamsters should be removed from the manuscript.

(9) Extended Data Figure 3C: Why are the normalization parameters different for testes (total reads) and MII oocytes (EERC spike-ins)?

(10) Figure 2B. The *Piwi2* mutant still shows some green staining expression that is similar to wild-type. More quantitative data is needed to determine whether this is real staining or background.

(11) Figure 4A and 4B: The authors state that "Despite the absence of *PIWIL3* 19-nt piRNAs, the distribution of small RNAs in maternal *Piwi3*^{-/-} embryos at 34 h.p.e.a. and 54 h.p.e.a. were more similar to those of WT embryos at 11 h.p.e.a. and 34 h.p.e.a., respectively." The figure does not support the statement; the length distributions of those wild-type stages do not look like those of the mutant.

Reviewer #3 (Remarks to the Author):

The manuscript by Lv et al. reports the characterization of all four PIWI genes and their associated piRNAs in golden hamsters with regard to their expression patterns and reproductive defects in their knockout-mutant golden hamsters. The immunofluorescence analysis of PIWIL1, 2 and 4 proteins indicated the evolutionarily conserved patterns of their expression and subcellular localization with regard to their orthologs in mice. The authors then generated *Piwil1*^{-/-}, *Piwil2*^{-/-}, *Piwil3*^{-/-}, and *Piwil4*^{-/-} golden hamsters, all of which showed normal viability without any discernible morphological or behavioral abnormalities. However, the *Piwil1*^{-/-} mutant was complete sterility in both males and females whereas *Piwil2*^{-/-} and *Piwil4*^{-/-} mutants were completely male sterile but did not show deduced female sterility. By contrast, *Piwil3*^{-/-} females displayed partial female fertility, as reported previously. Furthermore, *Piwil1*^{-/-} and *Piwil3*^{-/-} deficiency selectively impacted on the biogenesis of 29-nt and 19-nt piRNAs, respectively, with 29-nt piRNAs partially compensated for the loss of 19-nt piRNAs. This incomplete compensation was also reflected as maternal effect on embryogenesis, such that maternal *Piwil3*^{-/-} embryos were delayed in development but were not arrested at 2-cell and 4-cell stages. Finally, the authors systematically characterized piRNAs associated with the four PIWI proteins in wildtype and *Piwi* mutant testes at 3dpp and adulthood, which correlated these piRNAs roles in transposon silencing and gene expression.

This is a very systematic study of the PIWI-piRNA pathway in the golden hamster, a more fitting model than mice for investigating PIWI-piRNA functions in mammals and humans. The data are of very high quality and the conclusions are conservative and well justified. Although the most of the findings are expected and short of exciting novelty, this study, nevertheless, is a tour de force with substantial investment of research effort. There are a large number of findings that collectively complete the picture of the function of PIWI-piRNA pathway in the golden hamster. This paper will be well cited by PIWI-piRNA researchers and is very suitable for publication in Nature Communications. The manuscript is also well written. I did not spot any text that need to be edited. I recommend its publication without revision.

**Reviewer #1** (Remarks to the Author):

In this beautiful manuscript, the authors describe comprehensively the piRNA pathway
in golden hamsters. Using knockouts for all four PIWI proteins, they describe their
distribution in different developmental stages, sub-cellular localization, phenotypes
and small RNA populations. They compare their findings to the decades of information
available from study of mice, the other major mammalian model in the field. Recently
a couple of other publications have reported on the piRNA pathway in hamsters, but
this manuscript completes this series to encapsulate everything into one manuscript.

There are several key findings that are of interest to the field. The relevance of
piRNA pathway for the female germline (which is not the case in mice), compensation
between PIWI proteins during transposon silencing, role of PIWIL3 (a fourth protein
not found in mice), relevance of piRNA pathway for embryonic development (not see
in mice). This manuscript will be a reference for the community to access and cite. I
like the presentation of the figures- very clear and self-explanatory. In fact, one can
read the paper just by looking at the figures. I commend the authors for this heroic
effort in putting this manuscript together.

**RESPONSE:** We would like to express our sincere appreciation for the reviewer's
positive feedback regarding our study. It brings us great pleasure to hear that you found
our research informative and valuable to the field.

Minor suggestions.

1. The terms pr-piRNAs and pa-piRNAs are unnecessary. Just adds to the confusion.

**RESPONSE:** We thank the reviewer for the valuable suggestion. We have revised this
section and removed the terms 'pr-piRNAs' and 'pa-piRNAs' to avoid any potential
confusion.

2. The text constantly refers to mouse models while describing the results from hamster.
I think it is a bit confusing and distracting. Just describe from start to finish data from
hamsters. Only in the discussion, compare the two models.

**RESPONSE:** We thank the reviewer for this valuable suggestion. We have carefully

revised the manuscript and limited our comparison of the two models to the discussion
section in the new version of the paper.

3. There are a lot of repetitions in the text due to the mouse-hamster comparison.

**RESPONSE:** We thank the reviewer for pointing out this issue. We have moved
comparisons to the discussion section; this will hopefully reduce any tedium caused
by repetition.

**Reviewer #2** (Remarks to the Author):

Review uploaded as PDF.

Manuscript No.: 405166

“The non-redundant functions of PIWI family proteins in gametogenesis in golden
hamsters”

Xiaolong Lv, Wen Xiao, Yana Lai, Zhaozhen Zhang, Hongdao Zhang, Chen Qu, Li Hou,
Qin Chen, Duanduan Wang, Yun Gao, Yuanyuan Song, Xinjia Shui, Qinghua Chen,
Ruixin Qin, Shuang Liang, Wentao Zeng, Aimin Shi, Jianmin Li, Ligang Wu

Wu and colleagues have characterized the expression, localization, and loss-of-
function phenotypes for the four PIWI proteins in male and female golden hamsters.

The study design, the experiments, and results give convincing evidence of the
importance of the four PIWI proteins in gametogenesis in hamsters, but many of their
conclusions go too far beyond the existing evidence. Additionally, more detail is
required regarding sequencing and genetics to properly assess the study, and several
key experiments are missing statistical analysis or appear to have failed to correct for
multiple hypothesis testing.

**RESPONSE:** We would like to express our sincere gratitude to the reviewer for their
valuable and insightful comments. To address their concerns, we have conducted
additional experiments and provide more thorough analyses according to their
suggestions in the new version of the study. Furthermore, we have revised the text of
original manuscript based on their comments.

**Specific Concerns**

(1) How did the authors determine that PIWIL2 was present in IMC and PIWIL4 was
localized to the piP-body? What markers were used to identify mitochondria and piP
bodies? Please show the data that supports the colocalization of PIWIL2 and PIWIL4
with these subcellular structures in testis and of PIWIL3 with mitochondria in four-cell
embryos.

**RESPONSE:** As recommended, we have conducted new experiments and provide
evidence supporting the colocalization of PIWILs in the subcellular structures.
Specifically, we used protein markers including DCP1a and DDX6/p54 to detect piP
bodies in testes; TDRD1 and ATP5A to identify IMC in testes; and ATP5A to observe
mitochondria in four-cell embryos, as performed in previous studies (Ge et al., 2019,
*Mol Cell*; Hoop et al., 2021, *Mol Cell*). Our results show that PIWIL2 colocalizes with
DCP1A, TDRD1, and ATP5A in prospermatogonia, thus providing evidence of its
presence in piP bodies and IMC (Fig. s1B, s1E, and s1F). And PIWIL4 colocalize with
DCP1A and DDX6 in prospermatogonia, supporting its presence in piP bodies (Fig.
s1C and s1D). Furthermore, we found that PIWIL3 colocalizes with ATP5A in four-cell
embryos, indicating it can associate with mitochondria (Fig. s2G and s2H).

**Fig. s1 Expression and localization of PIWIs in male germ cells**

**(B-D)** Immunofluorescence staining of postnatal testes with anti-PIWIL2 and anti-DCP1A **(B)**,
 anti-PIWIL4 and anti-DCP1A **(C)**, or anti-PIWIL4 and anti-DDX6 **(D)** antibodies, respectively.
 Scale bar = 10 μm (top); Scale bar = 4 μm (bottom).

**(E-F)** Immunofluorescence staining of postnatal testes with anti-PIWIL2 and anti-TDRD1 **(E)** or
 anti-ATP5A **(F)** antibodies. Scale bar = 4 μm.

**Fig. s2 Expression and localization of PIWIs in female germ cells**

**(G)** Immunofluorescence staining of four-cell embryos with anti-PIWIL3 and anti-ATP5A
 antibodies to determine their colocalization. Scale bar, 4 μm.

**(H)** Fluorescence intensity profiles of PIWIL3 and ATP5A. Overlapping peaks indicate
 colocalization.

These new results are included in revised Fig. s1 and Fig. s2 and reported in the
corresponding Results section in the revised manuscript.

**(2)** For how many generations were the knockout hamsters backcrossed to eliminate
off-target CRISPR edits before the heterozygotes were crossed to obtain homozygotes
c?

**RESPONSE:** We thank the reviewer for highlighting this crucial but unintentionally
omitted information. To ensure the validity and reliability of our findings, all four Piwi
knockout hamsters were backcrossed for a minimum of nine generations before
crossing heterozygotes to obtain homozygotes. This information has been added to
the "Generation of Piwi mutant hamsters" subsection of the Methods in the following
statement: "All four Piwi mutant hamsters were backcrossed for a minimum of nine
generations before crossing heterozygotes to obtain homozygotes".

**(3)** How were small RNAs identified as piRNAs? For PIWIL1, 2, or 4, which are
methylated, a small RNA can be categorized as a piRNA if it is enriched in an oxidized
small RNA sequencing library or co-immunoprecipitated with the PIWI protein; for
PIWIL3, co-immunoprecipitation is the only way to identify piRNAs.

**RESPONSE:** We appreciate the reviewer's insightful question, and we agree that co-
immunoprecipitation (IP) or oxidization data are critical for identifying PIWI-specific
piRNAs

In our previous research (Zhang et al. 2021, *Nature cell biology*), we conducted
PIWIL1 and PIWIL3 IP and oxidation assays in wild-type and Piwil1 mutant oocytes or
testes (see Response Fig. 1, which is modified from Fig. 5b and 5c, Extended Data
Fig. 5f and 5g in the previous paper). These assays revealed the presence of three
groups of piRNAs in MII oocytes, including 23-nt and 29-30-nt oxidation resistant
piRNAs associated with PIWIL1 and 19-nt oxidation sensitive piRNAs associated with
PIWIL3.

In the current study, we conducted co-immunoprecipitation experiments with

antibodies specifically targeting PIWIL1, PIWIL2, or PIWIL4 in 3 d.p.p. or adult testes
 with small-RNA sequencing (see Fig. s10A, s10B, s11A, and s11E of the submitted
 manuscript). Our findings demonstrate that PIWIL2 binds piRNAs that peak at 27-nt in
 both adult and 3 d.p.p. testes, whereas PIWIL4 binds piRNAs that peak at 28-nt only
 in 3 d.p.p. testes and PIWIL1 binds piRNAs that peak at 29-30-nt only in adult testes.

In summary, we did use Co-IP with PIWIL1, PIWIL2, PIWIL3, and PIWIL4 to
 identify their specific, bound piRNAs, and also used oxidation to identify PIWIL1-
 piRNAs.

**Response Fig. 1 Identification of PIWIL1- and PIWIL3-piRNAs**

**(a)** The size distribution of PIWIL1- or PIWIL3-bound piRNAs identified by immunoprecipitation
 using anti-PIWIL1 or anti-PIWIL3 antibodies in WT and *Piwil1*^{m1/m1} MII oocytes, respectively.
 Rabbit non-specific immunoglobulin G (IgG) antibodies were used as a negative control. Two
 populations of piRNAs—22–24 nucleotides and 28–30 nucleotides—bound to PIWIL1; and 18–
 20-nucleotide piRNAs bound to PIWIL3. **(b)** The composition of small RNAs in WT and
 *Piwil1*^{m1/m1} MII oocytes with or without NaIO₄ treatment. **(c)** Composition of small RNAs in wild-

type and *Piwil1*^{m1/m1} testes with or without NaIO₄ oxidation treatment. The small RNA counts
were normalized by exogenous spike-in. **(d)** Size distribution of PIWIL1-bound piRNAs in wild-
type and *Piwil1*^{m1/m1} testes immunoprecipitated with PIWIL1-specific antibody. Rabbit non-
specific immunoglobulin G (IgG) served as a negative control. The small RNA counts were
normalized by exogenous spike-in.

**(4)** What supports the claim that additional 29 nt piRNAs are made in *Piwil3*^{-/-} ovaries
to compensate for loss of PIWIL3? Isn't it more likely that PIWIL3 and PIWIL1 compete
for a common set of pre-piRNAs and that in the absence of PIWIL3, these are loaded
into PIWIL1 and trimmed to 29 nt instead of 19 nt? Is the amount of PIWIL1 protein
greater in *Piwil3*^{-/-} than in wild-type?

**RESPONSE:** We sincerely appreciate this valuable suggestion from the reviewer and
apologize for any confusion caused by our explanation of the data. We intended to
convey that the increased levels of PIWIL1 29-nt piRNAs could functionally
compensate for the loss of PIWIL3-associated 19-nt piRNAs, not to suggest that
additional 29-nt piRNAs were produced through an unknown mechanism to
compensate for the loss of PIWIL3 production. We agree that PIWIL3 and PIWIL1
compete for a shared pool of pre-piRNAs, which aligns with our previous findings that
showed more than 70% of PIWIL3 piRNAs share identical 5' ends with PIWIL1 piRNAs.
Additionally, we concur that, in the absence of PIWIL3, PIWIL1 is loaded with and
processes more pre-piRNAs, leading to an increase in PIWIL1-piRNAs (Fig. 3A). As a
result, the increased levels of some PIWIL1 29-nt piRNAs might at least partially
compensate for the loss of PIWIL3 19-nt piRNA function in TE-repression, resulting in
the observed lack of change in TE expression due to PIWIL3 knockout (Fig. s5A).

As suggested, we have compared PIWIL1 protein expression levels in *Piwil3*
mutant oocytes with that of PIWIL3 levels in *Piwil1* mutant oocytes (Fig. s4E and s4F).
The results show that PIWIL3 protein levels are significantly decreased in the absence
of *Piwil1*, while PIWIL1 expression shows a non-significant, increasing trend in *Piwil3*
mutant oocytes. Thus, in the absence of PIWIL3, more pre-piRNAs are loaded in the
abundant PIWIL1 and trimmed to 29-nt instead of 19-nt. By contrast in *Piwil1* mutant

oocytes, impaired PIWIL3 expression results in lower efficiency processing of the
larger pool of pre-piRNAs, and consequently leading to almost unchanged expression
levels of 19-nt piRNAs (Fig. 3A).

To improve the clarity of our explanation of these data in the revised manuscript,
we have removed the sentence "suggesting widely compensatory production of 29-nt
piRNAs", and replaced "compensatory production of PIWIL1 29-nt piRNAs" with
"increased production of PIWIL1 29-nt piRNAs" (Line 220). We have also added
PIWIL1 and PIWIL3 protein expression data from WT and mutant oocytes. These new
results are shown in Fig. s4 of the revised manuscript and the text has been modified
accordingly (Line 202-213).

**Fig. s4 Female fertility phenotypes of *Piwi*-deficient golden hamsters**

**(E-F)** PIWIL3 protein levels in WT and *Piwil1*^{-/-} MII oocytes **(E)** and PIWIL1 levels in WT and
*Piwil3*^{-/-} mutant MII oocytes **(F)**. 17-20 oocytes from 4–6-month-old hamsters were collected
and lysed for protein analysis. Three biological replicates of oocytes were collected from each
of three WT, *Piwil1*^{-/-}, or *Piwil3*^{-/-} hamsters (n=2.7-3.3 oocytes/genotype). Signal intensity (area)
shows PIWIL1 or PIWIL3 protein accumulation detected by capillary electrophoresis Western
Blot assays. Significance was determined by unpaired two-tailed *t*-test. Data represent mean ±
187 s.e.m.

**(5)** The authors make two apparently contradictory claims: first, that in *Piwil1*^{-/-} germ
cells die before the pachytene stage and, second, that *Piwil1*^{-/-} testes are enriched
for spermatocytes arrested at the pachytene or diplotene stages. I don't understand
how both can be true.

**RESPONSE:** We are grateful to the reviewer for bringing point to our attention. In this
study, we quantified WT and *Piwil1* mutant spermatocytes by FACS using

Hoechst33342 and PI staining. Our analysis revealed that ~26% of the total
spermatocytes in *Piwil1* mutant testes were PI-positive, with the majority of these PI-
positive spermatocytes in the zygotene or pachytene stages (Fig. s9B). These results
suggest that a substantial proportion of spermatocytes die before reaching the
pachytene stage, while a significant proportion of them can survive and develop to the
pachytene or diplotene stage. However, these surviving pachytene or diplotene
spermatocytes are abnormal and exhibit anomalous staining of lectin peanut agglutinin
(PNA) around their nuclei, which we reported in our previous study (Zhang et al., 2021,
*Nat Cell Biol*). Moreover, few of these cells progress to the next stage and they
eventually detach from the tubules, flow into the cauda epididymis, and undergo cell
death (Fig. s9G).

To more clearly describe these observations, we have added the following
sentence to our results: "Although a significant proportion can survive to reach the
pachytene or diplotene stage, these spermatocytes were abnormal and few
progressed to the next stage; instead, they eventually detached from the tubules,
flowed into the cauda epididymis, and underwent cell death (Fig. s9G)." (Line 332-336).

**(6)** Given that the distribution of cell types in *Piwil1*^{-/-} testes differs from wild-type,
how can the authors be sure that the 1790 genes whose mRNA abundance increased
in the mutant testes is caused by a loss of piRNA-directed regulation rather than the
mismatch in tissue composition? To establish a change in mRNA abundance caused
by loss of piRNA function, purified germ cells must be compared to each other (e.g.,
FACS-purified primary spermatocytes from *Piwil1*^{-/-} vs. wild-type).

**RESPONSE:** We appreciate this valuable suggestion from the reviewer. As suggested,
we have purified pre-leptotene, leptotene/zygotene, pachytene, and diplotene
spermatocytes from WT and *Piwil1* mutant testes and conducted RNAseq to compare
transcriptomes at the same stage. Our results indicate that 4, 16, 32, and 115 genes
are dysregulated in the pre-leptotene, leptotene/zygotene, pachytene, and diplotene
*Piwil1* mutant spermatocytes, respectively (Fig. 6E). However, the number of
dysregulated TEs was negligible, suggesting that PIWIL1 primarily participates in gene

regulation rather than TE silencing.

Gene ontology (GO) analysis revealed that these differentially expressed genes
(DEGs) were enriched in functions related to “sperm motility” and “spermatid
development” (Fig. s12A), suggesting important roles in spermatocyte development
and maturation. These findings highlight the crucial transcriptional regulatory role of
PIWIL1 in spermatogenesis, even in leptotene and zygotene spermatocytes.

These new results are shown in new Fig. 6E and Fig. s12 of the revised
manuscript and the text has been modified accordingly (Line 391-404).

**Fig. 6 Impaired TE silencing or gene expression in *Piwi*-deficient spermatogenic cells**

**(E)** Analysis of differentially expressed consensus TEs and genes in purified pre-leptotene,
leptotene/zygotene, pachytene, and diplotene spermatocytes of WT and *Piwil1*^{-/-} testes.
TE/gene transcription levels were normalized to exogenous ERCC (External RNA Control
Consortium) RNA spike-in. Significantly up- or down-regulated TEs/genes (\geq four-fold; FDR <
0.01, permutation test) are indicated in red or blue, respectively. TEs and genes are indicated
by different shapes with TE or gene number shown at the top. Data are means of three or four
biological replicates.

**Fig. s12 Gene ontology (GO) analysis of DEGs in *Piwil1*^{-/-} spermatocytes**

**(A-C)** Top-ranking GO terms (biological processes) of differentially expressed genes in *Piwil1*-
 deficient leptotene/zygotene **(A)**, pachytene **(B)**, or diplotene **(C)** spermatocytes.

**(7)** The specific genes and transcription factors responsible for producing pachytene
 piRNA precursor RNAs have been defined in mice, macaque, and humans. Do the
 pachytene piRNAs in hamsters come from loci syntenic to those genes? From rodent-
 or hamster-specific loci?

**RESPONSE:** We thank the reviewer for this thoughtful question. We address this
 concern along with our response to questions 8, below.

**(8)** The claim that in hamsters there is no clear difference between pre-pachytene and
 pachytene piRNAs is semantics, not biology. Operationally, male mouse germ cells
 make transposon-silencing piRNAs, 3'-UTR-derived piRNAs, and "pachytene" piRNAs
 derived from a defined set of genes whose lncRNA transcripts are depleted of active
 transposons. For example, in male mice, the L1 transposon is desilenced in Miwi
 (*Piwil1*) mutants after the onset of meiosis. But the piRNAs involved do not derive from
 any of the pachytene piRNA loci and are instead derived from loci containing
 sequences antisense to L1. Despite the awkward nomenclature, "pachytene" piRNAs
 are defined by their source loci not their production at the pachytene stage of meiosis.
 The most evolutionarily conserved (by syteny not sequence) pachytene piRNA genes
 are also transcribed by a common set of transcription factors, A-MYB (MYBL1) and

TCFL5. Given that the pachytene piRNA genes responsible for the majority of human
“pachytene” piRNAs are found at syntenic locations in humans and are also
transcribed by A-MYB and TCFL5, it is highly unlikely that this class of piRNAs
(regardless of its name) is different in hamsters. Given the recent findings by Choi et
al. (PLoS Genet 2021) and Wu et al. (Nat Genet 2020) that the majority of pachytene
piRNAs have no detectable function in mice, it is unlikely that the composition and
functions of small RNAs obviously differ significantly among male placental mammals.
Similarly, the current manuscript provides little evidence that “the smRNA composition
and function in male germ cells are likely diverse in different mammals.”

**RESPONSE:** We sincerely thank the reviewer for helping to improve both our
understanding and our definition of so-called pachytene piRNAs. As the reviewer
suggested, we have identified the genomic loci responsible for generating pachytene
piRNA precursor RNAs using a previously described method (Özata et al., 2020,
*Nature Ecology&Evolution*). Briefly, we compared the abundance of piRNAs from same
genomic loci in 3 d.p.p. testes and adult testes. This analysis identified 97 loci
responsible for producing piRNAs with ≥ 4 -fold higher abundance in adult testes
compared to 3 d.p.p. testes, and were thus designated as pachytene piRNA loci (Fig.
7A). We then investigated whether these loci that produce pachytene piRNAs in golden
hamsters were syntenic with piRNA loci defined in mice and humans. Our results
indicate that ~58.8% (57/97) of pachytene piRNA-producing genes in the golden
hamster genome are found at syntenic locations in the mouse genome and ~24.7%
(24/97) are syntenic with corresponding piRNA loci in the human genome (Fig. 7A).
Approximately 38.1% (37/97) of them are from rodent-specific loci, and another ~33.0%
(32/97) are derived from golden hamster-specific loci. These findings suggest that a
substantial proportion of pachytene piRNA genes are syntenically conserved between
golden hamsters and other mammals.

As the reviewer mentioned, the most evolutionarily conserved pachytene piRNA
genes are often transcribed by a common set of transcription factors, A-MYB (MYBL1)
and TCFL5 (Li et al., 2013 Mol Cell; Yu et al., RNA, 2022). Intriguingly, we identified a
conserved A-MYB-binding motif in the promoter regions of the golden hamster

pachytene piRNA genes (Fig. 7B). These results support that the production of
 pachytene piRNA precursor RNAs is likely controlled by a specific set of transcription
 factors in golden hamster, which is consistent with reports in mice, macaques, and
 humans.

Next, we analyzed the expression patterns of pachytene piRNAs derived from the
 loci identified above in smRNAseq data from several developmental stages, which
 revealed that pachytene piRNAs were generated throughout the meiotic stage in
 golden hamsters (Fig. 7C). These findings sharply contrasted with the well-defined set
 of pachytene piRNAs produced in spermatocytes after entering the pachytene stage
 of meiosis in mice. These observations suggest that regulatory functions of pachytene
 piRNAs may initiate earlier in golden hamsters than those in mice, which is consistent
 with the phenotypic differences observed between *Piwil1*-deficient mice and *Piwil1*-
 deficient golden hamsters.

 **Fig. 7 Comparison of testicular pachytene piRNAs between golden hamsters and mice**
 **(A)** Heatmap of piRNA abundance for the golden hamster pachytene piRNA loci and the
 syntenic loci reported in mouse and human.
 **(B)** MEME identification of an A-MYB binding motif in the promoter regions of pachytene piRNA
 loci of golden hamsters.
 **(C)** Heatmap of piRNA abundance in the testes of golden hamster and mouse testes across
 developmental stages. In golden hamster, the 97 pachytene piRNA loci identified above are
 used with a corresponding set of well-defined pachytene piRNA loci from mice (Yu et al., 2021,
 *Nat Comm*). PL, pre-leptotene; L, leptotene; Z, zygotene; P, pachytene; RS, round sperm.

(D) Pie charts illustrating the genomic structural annotation of loci responsible for pachytene
piRNAs in mouse and golden hamster male germ cells, including intergenic, intronic, other
exonic, lncRNA, pseudogene, protein-coding, other repeats, DNA transposon, and retro
transposon regions.

Based on this analysis, we agree with the reviewer's assertion that pachytene
piRNAs in golden hamsters are similar to those in mice and humans. These piRNA
genes are primarily located in syntenically conserved regions and transcribed by a
common transcription factor. We also agree that the composition and function of small
RNAs in males are unlikely to differ significantly among placental mammals. We have
accordingly revised our manuscript to incorporate these points (Line 470-481).
However, we also observed that pachytene piRNA expression begins earlier in golden
hamsters than in mice, which may explain the greater severity of defects in
spermatogenesis in *Piwil1*^{-/-} golden hamsters. We have thus updated the Abstract to
more accurately reflect our conclusions, deleting the sentence "Notably, unlike mice,
none of the differences were found between piRNAs generated in pr-pachytene stage
and pachytene stage in golden hamsters". Moreover, we acknowledge the reviewer's
concerns regarding our previous statement that small RNA composition and function
in male germ cells varies significantly among mammalian species. We agree that our
evidence is insufficient to support this claim; we have rephrased the relevant text in the
Discussion section to present more accurate and defensible conclusions (Line 470-
493).

(9) The claim that the consequences of individual knockout of *Piwil1*, *Piwil2*, or *Piwil4*
are consistently more severe in golden hamsters than in mice may simply reflect the
use of one specific inbred strain (C57BL/6) for the mouse studies. This claim needs to
be softened.

**RESPONSE:** We thank the reviewer for bringing this point to our attention. We have
revised the sentence in question, which now reads as follows: "In particular, individual
knockout of *Piwil1*, *Piwil2*, or *Piwil4* leads to consistently more severe consequences

in golden hamsters than those observed in the C57BL/6 mouse model commonly used
in piRNA studies.” (Line 483-485).

**Important Data Concern**

Extended Data Figure 1 appears to be a composite of **multiple gels**. This type of
manipulation is unlikely to meet the journal standards. Please provide the uncropped
gels used to assemble the figure.

**RESPONSE:** We are grateful to the reviewer for bringing this matter to our attention.

As suggested, we now provide the uncropped gels in Source Data and we have

updated Fig. s1.

**Fig. s1 Expression and localization of PIWIs in male germ cells**

**(A)** Verification of antibody specificity for detection of PIWIL1, PIWIL2, PIWIL3, and PIWIL4 in
golden hamsters. Flag-tagged PIWIL1, PIWIL2, PIWIL3, and PIWIL4 were overexpressed in
293T cells, then immunoprecipitated using anti-PIWIL1, anti-PIWIL2, anti-PIWIL3, or anti-
PIWIL4 antibodies, respectively. Detection of the immunoprecipitated products using anti-Flag
antibody confirmed the specificity of anti-PIWI antibodies for their respective targets.

**Other Points**

**(1)** Figure 2A provides a far better demonstration of antibody specificity than Extended
Data Figure 1A. Could 2A be moved to Figure 1?

**RESPONSE:** We sincerely appreciate the valuable suggestion and fully acknowledge
that using mutant samples can provide a better demonstration of antibody specificity.
However, the *Piwi* mutants have not yet been introduced at this point in the manuscript,
and the rearrangement will thus introduce some logical problems with our study
narrative that are not easily rectified. Therefore, we respectfully maintain that these
antibodies will remain in Fig. 2A, and hopefully the reviewer will find that revised figure
S1A satisfies their concerns.

**(2)** In addition to reference 23, Loubalova et al. (Nat Cell Biol 2021) should be cited for
the female subfertility of *Piwil3* mutants.

**RESPONSE:** We thank the reviewer for pointing out this unintended oversight on our
part. This reference regarding the female subfertility of *Piwil3* mutants by Loubalova *et*
*al.* (Nat Cell Biol., 2021) is cited in the revised manuscript.

**(3)** Please provide the following information for the light microscopy methods:

- • Objective (magnification, N.A., immersion used, correction collar yes/no);
- • Filter sets and excitation sources and wavelengths; if known used power and excited
field of view;
- • Camera (manufacturer, model, pixel size and used settings);
- • Total system magnification and sampling rate of signal.

**RESPONSE:** We greatly appreciate the reviewer's attention to detail. We have

incorporated the recommended information regarding the light microscopy methods
into the Methods section.

**(4)** In Piwil3 mutant oocytes, what data show a geometric increase in 29 nt piRNAs
bound to PIWIL1? Why is the increase geometric?

**RESPONSE:** We are grateful to the reviewer for pointing out this poor choice of
wording. “Geometrically” has been replaced with “Obviously”.

(5) In Extended Data Figures 5A, 5E, and 6C were the data corrected for multiple
hypothesis testing. If not, the p-values need to be adjusted to take this into account.

**RESPONSE:** We sincerely appreciate the reviewer’s comment and we acknowledge
the importance of adjusting for multiple comparisons to mitigate the risk of type I error.
To address this concern, we have revised our analytical method for identifying
differentially expressed genes (DEGs) throughout the revised manuscript. We have
now performed differential expression analysis using a permutation test with the
Benjamini-Hochberg method for p value correction. Additionally, to reduce the
likelihood of false positives, we also adjusted the threshold for defining DEGs to a |fold
change| ≥ 4 and a p_{adj} value of < 0.01 . The details of our analytical approach have been
added to the Methods section of the revised manuscript. We have also updated the
images of Fig. 6B, Fig. 6E, Fig. 6F, and Fig. s5A accordingly.

**(6)** “Crosstalk” means the “unwanted transfer of signals between communication
channels.” The standard term for ping-pong between different PIWI proteins is
heterotypic ping-pong; ping-pong within a single type of PIWI protein is homotypic ping-
pong.

**RESPONSE:** We sincerely thank the reviewer for their advice. We have rectified this
incorrect language where necessary throughout the revised manuscript.

**(7)** Pachytene piRNAs in mice have been extensively characterized in both staged
whole testes (Li et al., Mol Cell 2013) and FACS purified cells (Gainetdinov et al., Mol

Cell 2018; Wu et al., Nat Genet 2020; Yu et al., RNA 2023), not simply in adult total
testes.

**RESPONSE:** We again thank the reviewer for their insight. As recommended, we
carefully reviewed the studies, obtained the sequencing data from public repositories
and conducted our own comprehensive analysis. Some of the results have been
incorporated into the revised manuscript, such as the synteny analysis among
pachytene piRNA loci and identification of the conserved transcription factor
responsible for producing pachytene piRNA precursor RNAs (Fig. 7), as discussed at
length above. Unfortunately, these available datasets only include FACS purified
pachytene and diplotene spermatocytes, and do not encompass FACS purified pre-
leptotene, leptotene, and zygotene spermatocytes. Therefore, these datasets cannot
be utilized for comparing the production of pachytene piRNAs in mice and golden
hamsters pre-pachytene spermatocytes.

**(8)** The report that “spermiRs” (not a term generally used by the field) are a very high
proportion of pre-meiotic small RNAs (ref. 39) is not consistent with other, more
quantitative analyses of spermatogonial miRNAs and piRNAs. Unless the authors can
further support these claims using publicly available sequencing data performed with
small RNA spike-ins and purified spermatogonia, the claim that there is a difference
between mice and hamsters should be removed from the manuscript.

**RESPONSE:** We thank the reviewer for raising this question. We analyzed the small
RNA sequencing data (Gainetdinov et al., Mol Cell, 2018) of spermatogonia and
compared the results with that of spermatogonial stem cells (SSCs) in reference 39
(Zhang et al., 2019, MBE). Our analysis revealed a non-trivial discrepancy in the ratio
of miRNAs to piRNAs between these two datasets (Response Fig. 2). This difference
might be attributable to several factors. First, spermatogonia include a broad range of
cells, each of which can also include a variety of subtypes, whereas SSCs are a
specific subtype of spermatogonia that display a unique capacity for self-renewal and
differentiation into specialized cell lineages. Second, the purity of sorted cells can
influence the ratio of miRNAs. For instance, a dispute persists regarding which cell

type, germ cells or Sertoli cells, expresses high levels of "Fragile-X miRNAs" (Fx-mir;
 also called miR-506 family, spermiR, or XmiR) (Ota et al., PLoS One, 2019; Zhang et
 al., Mol Biol Evol, 2019; Ramaiah et al., EMBO Rep, 2020; Wang et al., EMBO Rep).
 Finally, differences among library preparation methods for small RNA sequencing can
 also impact the miRNA to piRNA ratio. Sequence-fixed 5' or 3' adaptors can introduce
 ligation bias that may affect measurements of absolute abundance for different small
 RNAs (Giraldez et al., Nature Biotechnol, 2018). We noticed that a 3-N adaptor was
 used in Gainetdinov 's study, which might eliminate this effect.

**Response Fig. 2 Length distribution of small RNAs in spermatogonia or SSCs**

The small RNA counts were normalized to total mapped reads.

In our current study, we found that miRNAs are highly expressed in pre-pachytene
 mouse spermatocytes (~36-57% of total small RNAs), while miRNA expression is
 maintained at low levels in golden hamsters (<10%). As suggested by the reviewer, we
 searched almost every public resource to find available sequencing data generated
 with small RNA spike-ins and purified leptotene or zygotene spermatocytes.
 Unfortunately, these efforts were unsuccessful. In light of this absence of evidence, we
 have decided to remove this claim from the revised manuscript in order to ensure the
 rigor of our conclusions.

**(9) Extended Data Figure 3C: Why are the normalization parameters different for testes**
 **(total reads) and MII oocytes (EERC spike-ins)?**

**RESPONSE:** We thank the reviewer for bring this to our attention. We performed

RNAseq of testes in two batches, with each batch consisting of two biological replicates.
 Although ERCC spike-ins were included in the testis RNA libraries, the spike-in ratio
 was insufficient for normalization in one of the batches. Consequently, we normalized
 the expression of testis genes using the total mapped reads. Since the expression of
 individual *Piwi* genes was not compared between testes and oocytes, the use of
 different normalization parameters should not impact the results. However, to avoid
 any confusion and improve the transparency of our methods, we have divided panels
 C and D into four separate panels in Fig. s3 in the revised manuscript.

**Fig. s3 Generation and validation of *Piwi*-deficient golden hamsters**

**(C-D)** Validation of disrupted *Piwi* gene expression in *Piwi*-deficient testes **(C)** and MII oocytes
 **(D)** using RNA sequencing. Gene expression levels in testes are normalized to total mapped
 reads; gene expression levels in MII oocytes are normalized by ERCC spike-in. In box plots,
 the centre line represents the median value, the box borders represent the upper and lower
 quartiles (25th and 75th percentiles, respectively), and the ends of the top and bottom whiskers
 represent maximum and minimum scores, respectively. Data are means of the biological
 replicates for each mutant.

**(E-F)** Heatmaps of miRNA- and piRNA-related gene expression levels in WT and *Piwi*-deficient
 MII oocytes **(E)** and testes **(F)**. Gene expression levels are normalized by ERCC spike-in.

**(10)** Figure 2B. The *Piwi2* mutant still shows some green staining expression that is
 similar to wild-type. More quantitative data is needed to determine whether this is real
 staining or background.

**RESPONSE:** We thank the reviewer for their careful attention to detail. Although we
verified the specificity of anti-PIWIL2 antibody for immunostaining analysis in testes,
MII oocytes, and embryos, it should be noted that some low level of background
staining of this antibody could be observed in the nuclei of oocytes at the secondary
follicle stage. We found that a strong signal of PIWIL2 antibody was primarily detected
in the cytoplasm of quiescent oocytes in WT ovaries (Fig. 2B and Response Fig.2). In
contrast, no such staining was detected in *Piwil2*^{-/-} ovaries, indicating that PIWIL2
expression was indeed abolished in mutant oocytes. Therefore, the faint, diffuse signal
of PIWIL2 fluorescent antibody observed in secondary follicle stage mutant oocyte
nuclei cannot be considered a bona fide signal. Similarly, the apparent autofluorescent
staining observed in WT ovaries should also be considered background staining
artifact. To avoid potential confusion, we have added a note regarding this effect to the
figure legend of Fig. 2B in the revised manuscript.

**Response Fig. 2 Immunofluorescence detection of PIWIL2 expression in WT or *Piwil2*^{-/-}**
**ovaries.**

PIWIL2 staining is primarily observed in the cytoplasm of quiescent oocytes in WT ovaries,
while no staining is detected in *Piwil2*^{-/-} ovaries. The faint, diffuse signal in nuclei of secondary
follicle stage mutant oocytes is likely background autofluorescence. Scale bar = 200 μ m. Scale
517 bar (magnified inset) = 50 μ m.

**(11)** Figure 4A and 4B: The authors state that “Despite the absence of PIWIL3 19-nt
piRNAs, the distribution of small RNAs in maternal *Piwil3*^{-/-} embryos at 34 h.p.e.a.
and 54 h.p.e.a. were more similar to those of WT embryos at 11 h.p.e.a. and 34 h.p.e.a.,
respectively.” The figure does not support the statement; the length distributions of
those wild-type stages do not look like those of the mutant.

**RESPONSE:** We appreciate the reviewer pointing out this potential source of
confusion in our manuscript. Our results indicate that the decline in production of both
29-nt and 23-nt piRNAs is delayed in maternal *Piwil3*^{-/-} embryos compared to WT
embryos. Specifically, in WT embryos, the reduced production of 29-nt and 23-nt
piRNAs occurs at 34 h.p.e.a. and 54 h.p.e.a., respectively. In contrast, in maternal
*Piwil3*^{-/-} embryos, the decline in 29-nt piRNA production occurs at 54 h.p.e.a., while
production of 23-nt piRNAs begins after 54 h.p.e.a. To better explain these findings,
we have revised our description as follows: “In WT embryos, production of 29-nt
piRNAs ceases after 34 h.p.e.a., but ends after 54 h.p.e.a. in maternal *Piwil3*^{-/-} embryos
(Fig. 4A). Additionally, production of 23-nt piRNAs substantially decreases after 54
534 h.p.e.a. in WT embryos, but are still expressed at high levels comparable to WT
embryos at 34 h.p.e.a. at this stage in maternal *Piwil3*^{-/-} embryos (Fig. 4B). These
findings suggest a developmental delay in maternal *Piwil3*^{-/-} embryos compared to WT
embryos.” (Line 233-239).

**Reviewer #3** (Remarks to the Author):

The manuscript by Lv et al. reports the characterization of all four PIWI genes and their
associated piRNAs in golden hamsters with regard to their expression patterns and
reproductive defects in their knockout-mutant golden hamsters. The
immunofluorescence analysis of PIWIL1, 2 and 4 proteins indicated the evolutionarily
conserved patterns of their expression and subcellular localization with regard to their
orthologs in mice. The authors then generated *Piwil1*^{-/-}, *Piwil2*^{-/-}, *Piwil3*^{-/-}, and *Piwil4*^{-/-}
golden hamsters, all of which showed normal viability without any discernible
morphological or behavioral abnormalities. However, the *Piwil1*^{-/-} mutant was

complete sterility in both males and females whereas *Piwil2*^{-/-} and *Piwil4*^{-/-} mutants
were completely male sterile but did not show deduced female sterility. By contrast,
*Piwil3*^{-/-} females displayed partial female fertility, as reported previously. Furthermore,
*Piwil1*^{-/-} and *Piwil3*^{-/-} deficiency selectively impacted on the biogenesis of 29-nt and
19-nt piRNAs, respectively, with 29-nt piRNAs partially compensated for the loss of 19-
nt piRNAs. This incomplete compensation was also reflected as maternal effect on
embryogenesis, such that maternal *Piwil3*^{-/-} embryos were delayed in development
but were not arrested at 2-cell and 4-cell stages. Finally, the authors systematically
characterized piRNAs associated with the four PIWI proteins in wildtype and *Piwi*
mutant testes at 3dpp and adulthood, which correlated these piRNAs roles in
transposon silencing and gene expression.

This is a very systematic study of the PIWI-piRNA pathway in the golden hamster, a
more fitting model than mice for investigating PIWI-piRNA functions in mammals and
humans. The data are of very high quality and the conclusions are conservative and
well justified. Although the most of the findings are expected and short of exciting
novelty, this study, nevertheless, is a tour de force with substantial investment of
research effort. There are a large number of findings that collectively complete the
picture of the function of PIWI-piRNA pathway in the golden hamster. This paper will
be well cited by PIWI-piRNA researchers and is very suitable for publication in Nature
Communications. The manuscript is also well written. I did not spot any text that need
to be edited. I recommend its publication without revision.

**RESPONSE:** We sincerely appreciate the reviewer's supportive comments and their
acknowledgment of the significance of our study in the piRNA field. We hope our
research provides valuable and enduring insights that contribute to advancing this field.

We would like to take this opportunity to again express our heartfelt appreciation
to all the reviewers who have provided valuable and constructive comments regarding
our work. Their guidance has immensely helped to strengthen the technical rigor and

scientific significance of our manuscript. With these improvements, we hope that our
paper now meets the appropriately high standards required for publication in *Nature*
*Communications*.

REVIEWERS' COMMENTS

Reviewer #1 (Remarks to the Author):

The revised manuscript is now ready for publication. Good work by the authors.

Reviewer #2 (Remarks to the Author):

The authors have fully responded to my concerns. I am very much impressed by their willingness to invest considerable time and effort in gathering new data, performing additional analyses, and revising the text and figures of the manuscript. This is a lovely paper, and I strongly support publication without further delay.

**Reviewer #1 (Remarks to the Author):**

The revised manuscript is now ready for publication. Good work by the authors.

**Response:** We would like to express our sincere gratitude to the reviewer for
appreciating our work. Their recognition is truly valued and encourages us to continue
our efforts in conducting meaningful research.

**Reviewer #2 (Remarks to the Author):**

The authors have fully responded to my concerns. I am very much impressed by their
willingness to invest considerable time and effort in gathering new data, performing
additional analyses, and revising the text and figures of the manuscript. This is a lovely
paper, and I strongly support publication without further delay.

**Response:** We genuinely appreciate the reviewer's constructive comments, which
have greatly contributed to improving the quality of our manuscript, and recognition of
our efforts in addressing the issues raised.